# Urinary polycyclic aromatic hydrocarbon (PAH) metabolite concentrations in three pregnancy cohorts from 7 U.S. study sites

Erin E. Masterson[1], Anne M. Riederer[1]*, Christine T. Loftus[1], Erin R. Wallace[1], Adam A. Szpiro[2], Christopher D. Simpson[1], Revathi Muralidharan[1], Leonardo Trasande[3], Emily S. Barrett[4], Ruby H. N. Nguyen[5], Kurunthachalam Kannan[3], Morgan Robinson[3], Shanna Swan[6], W. Alex Mason[7], Nicole R. Bush[8,9], Sheela Sathyanarayana[1,10,11], Kaja Z. LeWinn[8], Catherine J. Karr[1,11]

1 Department of Environmental & Occupational Health Sciences, School of Public Health, University of Washington, Seattle, Washington, United States of America, 2 Department of Biostatistics, University of Washington, Seattle, Washington, United States of America, 3 Department of Pediatrics, Grossman School of Medicine, New York University, New York, New York, United States of America, 4 Rutgers School of Public Health, Environmental and Occupational Health Sciences Institute, Piscataway, New Jersey, United States of America, 5 Division of Epidemiology & Community Health, University of Minnesota School of Public Health, Minneapolis, Minnesota, United States of America, 6 Department of Environmental Medicine and Public Health, Icahn School of Medicine at Mount Sinai, New York, New York, United States of America, 7 Department of Preventive Medicine, University of Tennessee Health Science Center, Memphis, Tennessee, United States of America, 8 Department of Psychiatry and Behavioral Sciences, University of California at San Francisco School of Medicine, San Francisco, California, United States of America, 9 Department of Pediatrics, University of California at San Francisco School of Medicine, San Francisco, California, United States of America, 10 Seattle Children's Research Institute, Seattle, Washington, United States of America, 11 Department of Pediatrics, School of Medicine, University of Washington, Seattle, Washington, United States of America

* anneried@uw.edu

**Data Availability Statement:** A complete dataset corresponding to analyses in this paper cannot be made public because some data elements are

## Abstract

### Objective

Prenatal exposure to polycyclic aromatic hydrocarbons (PAHs) is associated with adverse birth and developmental outcomes in children. We aimed to describe prenatal PAH exposures in a large, multisite U.S. consortium.

### Methods

We measured 12 mono-hydroxylated metabolites (OH-PAHs) of 7 PAHs (naphthalene, fluorene, phenanthrene, pyrene, benzo(c)phenanthrene, chrysene, benz(a)anthracene) in mid-pregnancy urine of 1,892 pregnant individuals from the ECHO PATHWAYS consortium cohorts: CANDLE (n = 988; Memphis), TIDES (n = 664; Minneapolis, Rochester, San Francisco, Seattle) and GAPPS (n = 240; Seattle and Yakima, WA). We described concentrations of 8 OH-PAHs of non-smoking participants (n = 1,695) by site, socioeconomic characteristics, and pregnancy stage (we report intraclass correlation coefficients (ICC) for n = 677 TIDES participants).

owned by third parties, and because a careful review of IRB restrictions and data use agreements is required to avoid sharing that violates active legal agreements or results in release of sensitive data. These restrictions have been imposed by various IRBs and administrators at institutions that generated data. Please contact Lisa Younglove (lry@uw.edu) to request data access.

**Funding:** The ECHO PATHWAYS Consortium is funded by NIH UG3/UH3OD023271 (CK) (nih.gov). The Conditions Affecting Neurocognitive Development and Learning in Early Childhood (CANDLE) study was funded by the Urban Child Institute and NIH (R01 HL109977) (AM). The Infant Development and the Environment Study (TIDES) was funded by NIH (R01ES016863, R01 ES25169) (SS). This research was also supported by the University of Washington EDGE Center of the National Institutes of Health under award number: P30ES007033 (CK), and NIH P30ES005022 (EB). This research was also conducted using specimens and data collected and stored on behalf of the Global Alliance to Prevent Prematurity and Stillbirth (GAPPS) Repository. Dr. Kannan analyzed OH-PAHs in TIDES with support from the NYU ECHO Cohort Center (UG3/UH3OD023305 (LT), and for CANDLE and GAPPS with support of the ECHO PATHWAYS Consortium. The funders had no role in study design, data collection and analysis, decision to publish, or preparation of the manuscript.

**Competing interests:** The authors have declared that no competing interests exist.

## Results

Exposure to the selected PAHs was ubiquitous at all sites. 2-hydroxynaphthalene had the highest average concentrations at all sites. CANDLE had the highest average concentrations of most metabolites. Among non-smoking participants, we observed some patterns by income, education, and race but these were not consistent and varied by site and metabolite. ICCs of repeated OH-PAH measures from TIDES participants were $\leq 0.51$.

## Conclusion

In this geographically-diverse descriptive analysis of U.S. pregnancies, we observed ubiquitous exposure to low molecular weight PAHs, highlighting the importance of better understanding PAH sources and their pediatric health outcomes attributed to early life PAH exposure.

## Introduction

Polycyclic aromatic hydrocarbons (PAHs) are common toxicants that result from incomplete combustion of organic materials, including coal, crude oil, garbage, wood and gasoline. They occur in mixtures with various other toxic products of combustion. PAHs are found in ambient and indoor air, food, soil, and water, making human exposure possible via inhalation, ingestion, and skin contact [1,2]. Exposure occurs primarily via inhalation and diet [1] with key anthropogenic sources including tobacco smoke, vehicular emissions, high-temperature cooking and grilled/roasted/smoked foods, industrial food processing, coal power plants, asphalt-based products, and residential heating, including wood burning [1]. Natural sources include forest fires, volcanic activity, petroleum seeps, and coal deposits [1].

PAH exposure during pregnancy has been linked to a number of adverse birth and pediatric outcomes [3–5]. Because PAHs are generally metabolized within a few hours of exposure and mono-hydroxylated metabolites (OH-PAHs) are detectable in urine, urinary OH-PAH concentrations are frequently considered biomarkers of recent PAH exposure [6–8]. However, evidence shows that PAHs can accumulate in fatty tissue compartments such as fat and brain [9], thus urinary OH-PAH levels may, to a certain extent, reflect PAHs released from longer-term storage compartments in addition to recent exposures. In general, urinary PAH biomarkers are more convenient and cost-effective to collect and analyze compared to blood, thus they are increasingly being used in epidemiological studies of prenatal exposures and child health outcomes. To date, however, existing data on urinary OH-PAH levels in pregnancy are mostly derived from small studies representing a single geographic region, and/or databases that impute non-detectable observations using a single value, e.g., limit of detection (LOD)/square root of two, which can bias results where detection frequencies are low [10–13].

We built on these studies and described urinary OH-PAH levels among pregnant individuals from 7 sites in a large U.S. consortium, ECHO PATHWAYS, part of the Environmental Influences on Child Health Outcomes (ECHO) initiative of the U.S. National Institutes of Health (NIH) [14]. The ECHO PATHWAYS consortium provided harmonized prenatal exposure data, including urinary OH-PAHs. Maternal urinary concentrations of 8 OH-PAHs were measured in a sample of nearly 2,000 participants from 7 study sites residing in 6 different U.S. metropolitan areas. We compared our results to those of non-smoking females of reproductive age in the U.S. National Health and Nutrition Examination Survey (NHANES) 2011–2012.

We used maximum likelihood estimation (MLE) fitting methods to account for concentrations below the LOD [10–15]. Four study sites included repeated pregnancy urine samples, which enabled us to evaluate the temporal variability of OH-PAH concentrations across pregnancy.

## Materials and methods

### Participants

This analysis included pregnant participants in the three prospective pregnancy cohort studies comprising the ECHO PATHWAYS Consortium: Conditions Affecting Neurocognitive Development and Learning in Early childhood (CANDLE), The Infant Development and Environment Study (TIDES), and the Global Alliance to Prevent Prematurity and Stillbirth (GAPPS). Between 2007 and 2013 (and ongoing for GAPPS), participants were recruited from seven study sites across six U.S. metropolitan regions: Memphis (CANDLE), San Francisco, Minneapolis, Rochester, Seattle (TIDES), Seattle, Yakima, WA (GAPPS). The individual study designs, recruitment, consent and data collection procedures have been described elsewhere [16–18]. All participants provided written informed consent prior to enrolling in their respective cohort study and the larger PATHWAYS Consortium. All ECHO PATHWAYS Consortium research activities, including this analysis, were approved by the University of Washington Human Subjects Division. The authors of the present study did not have access to identifying information.

Inclusion criteria for the present analysis were urinary OH-PAH analysis during at least one time point in pregnancy, non-missing mid-pregnancy specific gravity (s.g.), and maternal prenatal smoking status data. The analytic sample included 1,892 participants overall: CANDLE (n = 988), TIDES (n = 664) and GAPPS (n = 240). S1 Fig shows the participant flow diagram by cohort and site. All participants included in this analysis contributed one urine sample from a mid-pregnancy visit. Among these participants, a subset of TIDES participants (n = 677) additionally provided at least one other urine sample from early and/or late pregnancy.

### OH-PAH measurements

Samples were stored at -80˚C in each cohort's respective biorepository until analysis for OH-PAHs at the New York State Department of Health, Wadsworth Laboratory. We used liquid-liquid extraction and LC-MS/MS to measure 12 mono-hydroxylated metabolites of seven PAH parent compounds (naphthalene, fluorene, phenanthrene, pyrene, benzo(c)phenanthrene, chrysene, benz[a]anthracene) in urine [19]. Briefly, samples (500 µL) were transferred into 15-mL glass tubes, spiked with 10 ng each of an isotopically labeled internal standard mixture, and mixed with 1 mL of 0.5 M ammonium acetate buffer containing 200 units/mL of β-glucuronidase/sulfatase enzyme (MP Biomedicals, LLC, Solon, OH, USA). Samples were gently mixed and incubated overnight at 37˚C, then diluted (2 mL HPLC water, followed by 7 mL of 80% pentane: 20% toluene, v:v) and shaken on a reciprocating shaker for one hour. After centrifugation, the supernatant was transferred to a new tube, concentrated under a gentle stream of nitrogen at 30˚C, and the final extract reconstituted with 250 µL of methanol.

Chromatographic separation was accomplished using a Waters Acquity I-Class UPLC system (Waters Corporation; Milford, MA, USA) connected with an Acquity UPLC BEH C18 column (50 × 2.1 mm, 1.7 µm, Waters; Milford, MA, USA). Identification and quantification of OH-PAHs was performed on an ABSCIEX 5500 Triple Quadrupole mass spectrometer (Applied Biosystems; Foster City, CA, USA) operated in electrospray ionization negative mode with multiple reaction monitoring. The mobile phases were methanol and water. The

peaks of 2-, 3-, 9-hydroxyfluorene and 1-, 9-hydroxyphenanthrene could not be chromatographically separated, so they are quantified as the sum of the individual metabolite concentrations.

Quality assurance/quality control (QA/QC) procedures included processing, for each batch of 100 samples, five duplicates of the following: method blank, matrix (urine) blank, and matrix (urine) spiked samples. Two duplicates of NIST SRM® (Standard Reference Material) 3672 (Organic Contaminants in Smokers' Urine) and SRM® 3673 (Organic Contaminants in Non-Smokers' Urine), containing certified values of several target OH-PAHs, were also processed. Synthetic urine (Cerilliant, Round Rock, Texas, USA) was used for the matrix blank and matrix spiked samples. HPLC water was used for sample/procedural blanks. Thirteen calibration standards were analyzed with each batch of samples, from which a quadratic calibration curve (0.02 ng/mL– 200 ng/mL) with a 1/x weighting was developed. Periodic injections of OH-PAH calibration standards were included throughout the sample run to ensure instrument stability in responses. Periodic instrumental blanks were also injected throughout the sample run to ensure no carryover or contamination.

For the TIDES cohort, s.g. of urine was determined using a handheld refractometer. For the CANDLE and GAPPS cohort data, s.g. of urine was determined using an Atago PAL-3 refractometer (Atago, Tokyo, Japan), which measures BRIX values. These values were further converted to specific gravity, according to the following equation: specific gravity = 1 + 0.004 X (BRIX%) [20,21].

**QC results.** Analyte-specific LODs were calculated as the lower value of three times the signal-to-noise ratio or the lowest calibration standard concentration. Samples with concentrations above the highest calibration standard were diluted and re-analyzed to fit the calibration range. Some target analytes were detected in the CANDLE and GAPPS method and matrix blanks; samples and matrix spike samples from these cohorts were blank corrected using the mean of the method blanks and matrix blanks, respectively. No analytes were detected in TIDES blanks, so those results were not blank corrected.

Overall, good analytical accuracy was achieved. Accuracy and precision results are included in S1 (CANDLE and GAPPS) and S2 (TIDES) Tables. Mean matrix spike recoveries (with TIDES corrected for internal standard recoveries) ranged from 92% (4-hydroxyphenanthrene) to 110% (1-hydroxychrysene) in CANDLE (n = 10), 69% (1-hydroxychrysene) to 116% (hydroxybenzo(c)phenanthrene) in TIDES (n = 26), and 83% (2/3/9-hydroxyfluorene) to 107% (1-hydroxychrysene) in GAPPS (n = 3). Mean recoveries of the certified concentrations of 1- and 2-hydroxynaphthalene, 2-, 3- and 4-hydroxyphenanthrene, and 1-hydroxypyrene in SRM 3673 (Non-Smokers' Urine) ranged from 65% to 123%. Mean recoveries of the certified concentrations of those analytes in SRM 3672 (Smokers' Urine) ranged from 80% to 122%. The standard deviations of the SRM replicates indicated good analytical precision (S1 and S2 Tables).

## Cohort characteristics

Maternal smoking during pregnancy and socioeconomic data (including annual household income at enrollment, education, maternal age at birth, race and ethnicity) were collected through questionnaires administered during pregnancy. We used the names of U.S. metropolitan areas to define the study sites in our description of the data although some participants resided outside of these areas. We relied on gestational age and date of urine sample collection to determine the pregnancy visit timing (e.g., early, mid, late). We classified participants as smokers during pregnancy if they self-reported any tobacco use during pregnancy or if their

urinary cotinine level was ≥ 200 ng/mL [22]. Cotinine analysis was completed using a method established for measuring perfluorochemicals and has been adapted for cotinine analysis [23].

## NHANES comparison

For comparison, we examined OH-PAH data from the U.S. National Health and Nutrition Examination Survey (NHANES) 2011–2012 [24]. Urinary OH-PAHs were measured in a sub-sample of NHANES participants aged 6 years and older using isotope dilution gas chromatography with tandem high-resolution mass spectrometry [24]. We restricted the NHANES data to non-smoking females aged 16–46 years (n = 535) to better reflect our PATHWAYS sample. We determined smoking status by self-report and urinary cotinine levels (≥ 200 ng/mL indicative of active smoking) [22]. Nineteen of these NHANES participants had a positive pregnancy test but we did not consider them separately because of the small sample size. We used the unweighted, uncorrected NHANES OH-PAH concentrations (ng/mL) for comparisons with PATHWAYS uncorrected OH-PAH concentrations (ng/mL).

## Statistical analyses

We described the analytic sample (N = 1,892) stratified by maternal smoking status during pregnancy given the clearly established association between tobacco smoke exposure and urinary OH-PAH concentrations and by study site [1]. We examined descriptive statistics by socioeconomic characteristics using arithmetic means and standard deviations for continuous measures and counts and proportions for categorical measures. We excluded 9 participants who were missing maternal smoking status during pregnancy data (cotinine or self-report) and another 9 participants who were missing specific gravity data. In the GAPPS cohort, cotinine was not measured for 62% of participants. Since only one GAPPS participant (among those with cotinine data) had cotinine ≥ 200 ng/mL, we assumed cotinine < 200 ng/mL for GAPPS participants with unmeasured cotinine data and relied on their self-report to classify smokers. We reported gestational age (in weeks) at urine sample collection for all visits by study site.

We evaluated the proportion of each site-specific sample with OH-PAH results > LOD. We reported uncorrected urine concentrations and those corrected for s.g.: (OH-PAH conc.* [(median s.g.)-1/(s.g.-1)], where s.g. represents the s.g. of the urine sample) to account for variation in dilution of the urinary samples. All concentrations are reported in ng/mL. We calculated descriptive statistics for both uncorrected and s.g.-corrected OH-PAH concentrations by study site and maternal smoking status during pregnancy. Analyses were conducted based on a complete case analysis approach; in the case of missing data, sub-sample sizes were noted in the Results tables and figures.

We characterized OH-PAH concentrations by study site and socioeconomic characteristics for non-smoking participants only (n = 1,695). For TIDES participants with multiple pregnancy urine samples, we also characterized OH-PAHs by stage of pregnancy (n = 677 TIDES participants) and estimated intraclass correlation coefficients (ICC) by study site and in the pooled TIDES sample for metabolites with >95% detection frequencies at all four TIDES sites (2-hydroxynaphthalene, 2- and 3-hydroxyphenanthrene) using the PROC MIXED command in SAS. To reduce bias and account for left-censoring in the OH-PAH data, we used a maximum likelihood estimation (MLE) method and the PROC LIFEREG command in SAS or the R 'fitdistrplus' package to fit lognormal distributions and estimate geometric means (GM) and geometric standard deviations (GSD) of s.g.-corrected OH-PAH values [10–12,15,25–28]. We visually compared each MLE-fitted distribution with the corresponding histogram of the log-transformed OH-PAH values as well as the mode, which we estimated using the parzen

function in the R 'modeest' package [29]. S2 and S3 Figs show examples of MLE fitted distributions, histograms and modes for analytes with relatively low (i.e., ~10%) and high (i.e., ~35%), respectively, degrees of left-censoring. In our Results figures, we indicated our confidence in the MLE fit and central tendency statistics using regular font (interpret with caution: detection frequency between 60 and 85%) and bold font (very confident: detection frequency > 85%). We did not report central tendency statistics when site-specific detection frequencies were < 60% [30]. We used the R clikcorr package, which also uses MLE to account for left-censored observations, to estimate correlation coefficients between log-transformed individual OH-PAH metabolites within sites [31].

Since income and education were previously associated with urinary OH-PAH concentrations in pregnant [32] and non-pregnant people [33–34], we compared GM/GSD OH-PAH concentrations across income and education categories. We also used Tobit regression to formally evaluate associations between income and education and log-transformed OH-PAH concentrations, accounting for left censoring in the data and adjusting for urinary specific gravity. In these regressions, income was modeled following the recommended approach for multi-site PATHWAYS analyses—harmonized household size (2–3, 4, 5, ≥ 6 people) and income (adjusted for region and inflation to 2012 $USD) from the 4–6 year old visit was used instead of the prenatal visit because of the proportion of right-censored prenatal TIDES and GAPPS observations, which was addressed at the 4–6 year old visit with additional upper income categories. Natural log household income interacted with household size ('household size adjusted household income') and the corresponding main effects were included in the models. For the education regressions, the "< High school" and "High school completion" categories were combined to avoid low cell counts in the "< High school" category. Complete case analysis was used for the Tobit regressions. We also explored associations with maternal age (years), race (White, Black/African American, and an "Other" category comprised of Asian, Native Hawaiian/Pacific Islander, American Indian/Alaska Native, Other, and Multiple to avoid low cell counts), and ethnicity (Not Hispanic/Latino, Hispanic/Latino) using the same approach.

Analyses were conducted in SAS version 9.4 (SAS Institute, Cary, NC, USA) and R version 4.0.0 (R Foundation for Statistical Computing, Vienna, Austria).

## Results

### Study sample description

Our final analytic sample included 1,892 pregnant participants, aged 16–49 years, from seven study sites across six U.S. metropolitan regions: Memphis, TN; San Francisco, CA; Minneapolis, MN; Rochester, NY; Seattle, WA (2 distinct samples: TIDES and GAPPS); and Yakima, WA (Table 1). The mean age was 29 (SD: 6) years, with variation across sites; in general, CANDLE, TIDES Rochester, and GAPPS Yakima participants were younger than participants at other sites. Household income also differed across sites. Annual household incomes within the Memphis and Rochester sites tended to be lower than at the other five sites. Overall, most of the study sample completed at least high school (91%) and over half (57%) also completed college or technical school, but this too varied by site. The study sample was predominantly White (51%) and Black/African American (38%), however race was not evenly distributed across the sites. Most of the Black/African American participants resided at the Memphis and Rochester study sites.

All participants provided urine samples during the mid-pregnancy visit, most of which were collected during the second trimester (defined as 14 weeks and 0 days to 27 weeks and 6 days) [35]. The median gestational age at the mid-pregnancy urine sample collection was 22

**Table 1. Description of maternal smoking status during pregnancy & socioeconomic characteristics, overall and by study site (N = 1,892).**

| Characteristic | Overall N = 1,892 | CANDLE | | TIDES | | | GAPPS | |
|---|---|---|---|---|---|---|---|---|
| | | Memphis n = 988 | San Francisco n = 169 | Minneapolis n = 167 | Rochester n = 190 | Seattle n = 138 | Seattle n = 112 | Yakima n = 128 |
| **Smoking status during pregnancy [a]** | | | | | | | | |
| non-smoker | 1,695 (90) | 885 (90) | 167 (99) | 163 (98) | 145 (76) | 135 (98) | 91 (81) | 109 (85) |
| smoker | 197 (10) | 103 (10) | 2 (1) | 4 (2) | 45 (24) | 3 (2) | 21 (19) | 19 (15) |
| **Maternal age at delivery** | | | | | | | | |
| missing data, n | 31 | 20 | 8 | 1 | 0 | 0 | 0 | 2 |
| < 24 years | 415 (22) | 321 (32) | 1 (1) | 6 (4) | 62 (33) | 5 (4) | 4 (4) | 15 (12) |
| 24–29 years | 565 (30) | 331 (34) | 14 (8) | 52 (31) | 65 (34) | 34 (25) | 23 (21) | 46 (36) |
| 30–33 years | 444 (23) | 185 (19) | 43 (25) | 56 (34) | 39 (21) | 49 (36) | 34 (30) | 38 (30) |
| > 33 years | 438 (23) | 131 (13) | 103 (61) | 52 (31) | 24 (13) | 50 (36) | 51 (46) | 27 (21) |
| **Annual household income (USD, k = 1,000)** | | | | | | | | |
| missing data, n | 105 | 73 | 0 | 4 | 10 | 5 | 7 | 6 |
| < 15k (CANDLE, TIDES) | 379 (20) | 257 (26) | 6 (4) | 12 (7) | 70 (37) | 11 (8) | – | – |
| < 20k (GAPPS) | – | – | – | – | – | – | 9 (8) | 14 (11) |
| 15k - 25k(CANDLE, TIDES) | 211 (11) | 144 (15) | 3 (2) | 13 (8) | 30 (16) | 4 (3) | – | – |
| 20–30k (GAPPS) | – | – | – | – | – | – | 7 (6) | 10 (8) |
| 25k–45k(CANDLE, TIDES) | 262 (14) | 170 (17) | 6 (4) | 15 (9) | 29 (15) | 13 (9) | – | – |
| $30–40k (GAPPS) | – | – | – | – | – | – | 6 (5) | 9 (7) |
| $40–50k (GAPPS) | – | – | – | – | – | – | 3 (3) | 11 (9) |
| 45k–55k(CANDLE, TIDES) | 131 (7) | 76 (8) | 5 (3) | 16 (10) | 7 (4) | 9 (7) | – | – |
| 50–60k (GAPPS) | – | – | – | – | – | – | 5 (4) | 13 (10) |
| 55k–65k(CANDLE, TIDES) | 107 (6) | 54 (6) | 6 (4) | 12 (7) | 7 (4) | 6 (4) | – | – |
| 60k-70k (GAPPS) | – | – | – | – | – | – | 1 (1) | 21 (16) |
| 65k–75k(CANDLE, TIDES) | 113 (6) | 58 (6) | 7 (4) | 16 (10) | 6 (3) | 15 (11) | – | – |
| 70k–80 (GAPPS) | – | – | – | – | – | – | 7 (6) | 4 (3) |
| ≥ 75k (CANDLE, TIDES) | 488 (26) | 156 (16) | 136 (80) | 79 (47) | 31 (16) | 75 (54) | – | – |
| ≥ 80k (GAPPS) | – | – | – | – | – | – | 67 (60) | 40 (31) |
| **Maternal highest level of education completed** | | | | | | | | |
| missing data, n | 6 | 1 | 0 | 3 | 0 | 1 | 1 | 0 |
| < High school | 164 (9) | 103 (10) | 2 (1) | 4 (2) | 47 (25) | 1 (1) | 1 (1) | 6 (5) |
| High school completion | 645 (34) | 468 (47) | 13 (8) | 15 (9) | 75 (39) | 12 (9) | 18 (16) | 44 (34) |
| Graduated college or technical school | 614 (32) | 296 (30) | 53 (31) | 62 (37) | 38 (20) | 57 (41) | 55 (49) | 53 (41) |
| Some graduate work or grad/ professional degree | 463 (24) | 120 (12) | 101 (60) | 83 (50) | 30 (16) | 67 (49) | 37 (33) | 25 (20) |
| **Maternal race** | | | | | | | | |
| missing data, n | 17 | 0 | 0 | 4 | 4 | 2 | 3 | 4 |
| White | 955 (51) | 297 (30) | 128 (76) | 134 (80) | 91 (48) | 114 (83) | 87 (78) | 104 (81) |
| Black/African American | 717 (38) | 631 (64) | 3 (2) | 12 (7) | 64 (34) | 3 (2) | 4 (4) | 0 (0) |
| Asian | 53 (3) | 9 (1) | 21 (12) | 7 (4) | 2 (1) | 7 (5) | 7 (6) | 0 (0) |
| Other or multiple | 150 (8) | 51 (5) | 17 (10) | 10 (6) | 29 (15) | 12 (9) | 11 (10) | 20 (16) |
| **Maternal ethnicity** | | | | | | | | |
| missing data, n | 8 | 0 | 0 | 5 | 0 | 1 | 2 | 0 |
| Not Hispanic or Latino | 1,772 (94) | 971 (98) | 149 (88) | 158 (95) | 162 (85) | 127 (92) | 99 (88) | 106 (83) |
| Hispanic/Latino | 112 (6) | 17 (2) | 20 (12) | 4 (2) | 28 (15) | 10 (7) | 11 (10) | 22 (17) |
| **Gestational age (weeks) at urine sample collection** | | | | | | | | |

*(Continued)*

**Table 1.** (Continued)

| Characteristic | Overall N = 1,892 | CANDLE | TIDES | | | | GAPPS | |
| --- | --- | --- | --- | --- | --- | --- | --- | --- |
| | | Memphis n = 988 | San Francisco n = 169 | Minneapolis n = 167 | Rochester n = 190 | Seattle n = 138 | Seattle n = 112 | Yakima n = 128 |
| Early pregnancy, n | 649 | – | 169 | 167 | 176 | 137 | – | – |
| mean ± SD | 10.87 ± 2.09 | – | 12.20 ± 1.30 | 10.11 ± 2.24 | 10.34 ± 2.14 | 10.84 ± 1.86 | – | – |
| range | 5.1, 23.3 | – | 6.3, 14.1 | 6.1, 20.4 | 5.1, 23.3 | 6.3, 13.9 | – | – |
| median | 11.1 | – | 12.4 | 10.40 | 10.3 | 11.4 | – | – |
| Mid pregnancy, n | 1,867 | 963 | 169 | 167 | 190 | 138 | 112 | 128 |
| mean ± SD | 22.07 ± 3.50 | 22.96 ± 3.05 | 19.16 ± 1.49 | 19.93 ± 2.22 | 19.35 ± 3.02 | 25.66 ± 3.71 | 21.28 ± 4.86 | 22.12 ± 4.78 |
| range | 12.9, 35.1 | 15.3, 29.6 | 14.6, 25.1 | 14.0, 29.3 | 12.9, 30.9 | 17.1, 35.1 | 13.1, 39.3 | 9.3, 36.3 |
| median | 21.4 | 22.9 | 19.30 | 19.70 | 19.20 | 25.95 | 21.3 | 24.1 |
| Late pregnancy, n | 617 | – | 164 | 162 | 174 | 117 | – | – |
| mean ± SD | 32.10 ± 3.05 | – | 32.52 ± 2.39 | 31.78 ± 2.11 | 31.28 ± 2.58 | 36.68 ± 2.27 | – | – |
| range | 25.7, 41.1 | – | 28.3, 40.0 | 25.7, 38.0 | 26.0, 39.3 | 29.7, 41.1 | – | – |
| median | 32.10 | – | 32.30 | 31.30 | 31.00 | 37.00 | – | – |

[a]serum cotinine ≥ 200 ng/mL or maternal self-report of smoking during pregnancy.

weeks (range: 13–35 weeks). Median gestational age of mid-pregnancy urine collection differed across sites (overall range: 19–25 weeks) (Table 1). Mid-pregnancy samples tended to be collected at a slightly earlier gestational age in the San Francisco, Minneapolis and Rochester study sites and later at the Memphis and Seattle study sites.

Overall, most participants did not smoke during their pregnancy (90%). However, non-smoking status varied among study sites, from 76–99%, with some smoking during pregnancy occurring at the Memphis, Rochester, Seattle (GAPPS), Yakima sites and almost none at the San Francisco, Minneapolis and Seattle (TIDES) sites.

## OH-PAH results

**OH-PAH concentrations.** Table 2 shows OH-PAH results by study site and participant smoking status during pregnancy, and uncorrected NHANES 2011–2012 OH-PAH concentrations for females of reproductive age for comparison. We did not report data for smokers at the following TIDES sites due to small sub-sample sizes: San Francisco (n = 2), Minneapolis (n = 4), Seattle (n = 3). Among the four study sites for which we reported OH-PAH concentrations stratified by smoking status, we observed higher median concentrations among smokers compared to non-smokers for 1- and 2-hydroxynaphthalene, 2/3/9-hydroxyfluorene, and 1-hydroxypyrene in the Memphis and Rochester study samples. These differences were less notable and inconsistent for the two smaller GAPPS samples. Differences in median concentrations between smokers and non-smokers were less notable for the phenanthrene metabolites at all study sites.

We detected 2-hydroxynaphthalene in nearly all samples, regardless of site or smoking status, and 2-hydroxynaphthalene detection frequencies were high in all sites. Detection frequencies were also high (> 70%) for 1-hydroxynaphthalene, 1/9-hydroxyphenanthrene, and 2- and 3-hydroxyphenanthrene, with some site differences possibly explained by differences in LODs and/or differences in exposures. The higher LOD for 2/3/9-hydroxyfluorene in the TIDES cohort sites compared to other cohorts likely explains the lower detection frequencies for this analyte, while detection frequencies for 4-hydroxyphenanthrene were < 70% at most sites despite the similar LODs. Despite sensitive LODs, 1-hydroxypyrene detection frequencies

**Table 2. Urinary mono-hydroxylated PAH (OH-PAHs) concentrations from mid pregnancy visit by study site and prenatal smoking status (N = 1,892).**

| Analyte (abbreviation) | Cohort, site | Smoking status | LOD (ng/mL) | Percent detected (%) | OH-PAH concentrations (uncorrected) (ng/mL) | | | | s.g.-corrected OH-PAH concentrations (ng/mL) | | | |
|---|---|---|---|---|---|---|---|---|---|---|---|---|
| | | | | | Min | Median | 75th perc | Max | Min | Median | 75th perc | Max |
| **1-hydroxynaphthalene (1-OH-NAP)** | | | | | | | | | | | | |
| | CANDLE, Memphis | non-smokers (n = 885) | 0.020 | 100 | 0.04 | 0.89 | 2.12 | 331.00 | 0.07 | 1.03 | 2.14 | 331.00 |
| | | smokers (n = 103) | | 100 | 0.07 | 2.95 | 6.67 | 54.33 | 0.10 | 3.43 | 7.91 | 36.95 |
| | TIDES, San Francisco | non-smokers (n = 167[a]) | 0.04 | 81 | <LOD | 0.36 | 0.90 | 33.70 | <LOD | 0.55 | 1.13 | 102.60 |
| | TIDES, Minneapolis | non-smokers (n = 163[a]) | 0.04 | 59 | <LOD | 0.12 | 0.37 | 5.60 | <LOD | 0.16 | 0.44 | 10.96 |
| | TIDES, Rochester | non-smokers (n = 145) | 0.04 | 59 | <LOD | 0.13 | 0.54 | 109.50 | <LOD | 0.16 | 0.46 | 63.20 |
| | | smokers (n = 45) | | 84 | <LOD | 1.92 | 3.17 | 28.15 | <LOD | 1.76 | 3.35 | 63.22 |
| | TIDES, Seattle | non-smokers (n = 135[a]) | 0.04 | 67 | <LOD | 0.18 | 0.47 | 86.00 | <LOD | 0.25 | 0.53 | 58.91 |
| | GAPPS, Seattle | non-smokers (n = 91) | 0.017 | 100 | 0.02 | 0.31 | 0.65 | 2.26 | 0.03 | 0.33 | 0.59 | 2.54 |
| | | smokers (n = 21) | | 100 | 0.05 | 0.35 | 1.42 | 12.14 | 0.14 | 0.69 | 1.26 | 7.17 |
| | GAPPS, Yakima | non-smokers (n = 109) | 0.017 | 97 | <LOD | 0.51 | 0.96 | 19.78 | <LOD | 0.47 | 0.78 | 19.62 |
| | | smokers (n = 19) | | 95 | <LOD | 0.38 | 0.74 | 2.16 | <LOD | 0.43 | 0.66 | 1.33 |
| | NHANES 2011-2012[b] | non-smokers (n = 439) | 0.044 | 100 | 0.05 | 0.90 | 2.01 | 80 | – | – | – | – |
| | | smokers (n = 98) | | 100 | 0.19 | 10 | 18 | 260 | – | – | – | – |
| **2-hydroxynaphthalene (2-OH-NAP)** | | | | | | | | | | | | |
| | CANDLE, Memphis | non-smokers (n = 885) | 0.025 | 99 | <LOD | 4.53 | 8.44 | 227.99 | <LOD | 4.93 | 7.97 | 136.80 |
| | | smokers (n = 103) | | 100 | 0.83 | 8.45 | 18.10 | 117.00 | 1.11 | 9.71 | 17.82 | 91.57 |
| | TIDES, San Francisco | non-smokers (n = 167[a]) | 0.017 | 100 | <LOD | 1.87 | 5.11 | 33.20 | <LOD | 2.53 | 5.05 | 37.90 |
| | TIDES, Minneapolis | non-smokers (n = 163[a]) | 0.017 | 99 | <LOD | 1.72 | 3.62 | 82.00 | <LOD | 2.11 | 3.78 | 74.89 |
| | TIDES, Rochester | non-smokers (n = 145) | 0.017 | 99 | <LOD | 3.52 | 8.05 | 90.50 | <LOD | 2.96 | 6.14 | 56.36 |
| | | smokers (n = 45) | | 98 | <LOD | 10.20 | 16.60 | 110.00 | <LOD | 8.96 | 16.07 | 52.88 |
| | TIDES, Seattle | non-smokers (n = 135[a]) | 0.017 | 100 | 0.07 | 1.88 | 4.76 | 24.30 | 0.03 | 2.51 | 4.27 | 26.18 |
| | GAPPS, Seattle | non-smokers (n = 91) | 0.018 | 100 | 0.14 | 2.20 | 4.44 | 44.20 | 0.57 | 2.37 | 4.97 | 31.69 |
| | | smokers (n = 21) | | 100 | 0.28 | 1.28 | 4.70 | 22.03 | 0.78 | 2.05 | 4.70 | 13.72 |
| | GAPPS, Yakima | non-smokers (n = 109) | 0.018 | 100 | 0.08 | 3.94 | 7.80 | 94.50 | 0.37 | 3.61 | 6.97 | 39.90 |
| | | smokers (n = 19) | | 100 | 0.63 | 3.33 | 4.44 | 8.67 | 0.79 | 3.43 | 4.52 | 8.56 |
| | NHANES 2011-2012[b] | non-smokers (n = 439) | 0.042 | 100 | 0.14 | 4.9 | 11 | 180 | – | – | – | – |
| | | smokers (n = 98) | | 100 | 1.5 | 15 | 22 | 51 | – | – | – | – |
| **2-hydroxyfluorene/3-hydroxyfluorene/9-hydroxyfluorene (2/3/9-OH-FLUO)** | | | | | | | | | | | | |
| | CANDLE, Memphis | non-smokers (n = 885) | 0.120 | 97 | <LOD | 0.86 | 1.53 | 47.1 | <LOD | 0.90 | 1.46 | 29.23 |
| | | smokers (n = 103) | | 100 | 0.21 | 3.02 | 7.06 | 29.3 | 0.43 | 3.54 | 6.88 | 35.17 |

*(Continued)*

**Table 2.** (Continued)

| Analyte (abbreviation) | Cohort, site | Smoking status | LOD (ng/mL) | Percent detected (%) | OH-PAH concentrations (uncorrected) (ng/mL) | | | | s.g.-corrected OH-PAH concentrations (ng/mL) | | | |
|---|---|---|---|---|---|---|---|---|---|---|---|---|
| | | | | | Min | Median | 75th perc | Max | Min | Median | 75th perc | Max |
| TIDES, San Francisco | | non-smokers (n = 167[a]) | 0.48 | 35 | <LOD | <LOD | 0.63 | 3.71 | <LOD | <LOD | 0.99 | 4.82 |
| TIDES, Minneapolis | | non-smokers (n = 163[a]) | 0.48 | 18 | <LOD | <LOD | <LOD | 5.80 | <LOD | <LOD | <LOD | 13.24 |
| TIDES, Rochester | | non-smokers (n = 145) | 0.48 | 50 | <LOD | <LOD | 1.10 | 6.25 | <LOD | <LOD | 1.07 | 2.80 |
| | | smokers (n = 45) | | 80 | <LOD | 2.29 | 6.15 | 21.25 | <LOD | 2.82 | 4.68 | 12.13 |
| TIDES, Seattle | | non-smokers (n = 135[a]) | 0.48 | 23 | <LOD | <LOD | <LOD | 177.50 | <LOD | <LOD | <LOD | 148.74 |
| GAPPS, Seattle | | non-smokers (n = 91) | 0.017 | 95 | <LOD | 0.10 | 0.18 | 0.75 | <LOD | 0.11 | 0.18 | 0.50 |
| | | smokers (n = 21) | | 91 | <LOD | 0.14 | 0.30 | 7.56 | <LOD | 0.16 | 0.22 | 3.94 |
| GAPPS, Yakima | | non-smokers (n = 109) | 0.017 | 97 | <LOD | 0.16 | 0.27 | 76.62 | <LOD | 0.15 | 0.23 | 45.49 |
| | | smokers (n = 19) | | 100 | 0.02 | 0.14 | 0.23 | 0.50 | 0.06 | 0.16 | 0.20 | 0.40 |
| NHANES 2011-2012[b] | | non-smokers (n = 438) | 0.010 | 95 | <LOD | 0.48 | 0.87 | 6.6 | – | – | – | – |
| | | smokers (n = 98) | | 100 | 0.13 | 2.8 | 4.4 | 18 | – | – | – | – |
| **1-hydroxyphenanthrene/9-hydroxyphenanthrene (1/9-OH-PHEN)** | | | | | | | | | | | | |
| CANDLE, Memphis | | non-smokers (n = 885) | 0.080 | 83 | <LOD | 0.30 | 0.56 | 18.39 | <LOD | 0.34 | 0.57 | 11.41 |
| | | smokers (n = 103) | | 85 | <LOD | 0.46 | 0.82 | 3.08 | <LOD | 0.49 | 0.82 | 2.48 |
| TIDES, San Francisco | | non-smokers (n = 167[a]) | 0.007 | 92 | <LOD | 0.09 | 0.20 | 1.42 | <LOD | 0.13 | 0.22 | 1.14 |
| TIDES, Minneapolis | | non-smokers (n = 163[a]) | 0.007 | 82 | <LOD | 0.08 | 0.19 | 1.59 | <LOD | 0.09 | 0.18 | 1.64 |
| TIDES, Rochester | | non-smokers (n = 145) | 0.007 | 94 | <LOD | 0.16 | 0.31 | 2.13 | <LOD | 0.12 | 0.23 | 4.86 |
| | | smokers (n = 45) | | 96 | <LOD | 0.21 | 0.39 | 1.21 | <LOD | 0.21 | 0.35 | 0.69 |
| TIDES, Seattle | | non-smokers (n = 135[a]) | 0.007 | 90 | <LOD | 0.06 | 0.14 | 2.23 | <LOD | 0.06 | 0.11 | 1.13 |
| GAPPS, Seattle | | non-smokers (n = 91) | 0.017 | 93 | <LOD | 0.07 | 0.13 | 0.53 | <LOD | 0.07 | 0.13 | 0.83 |
| | | smokers (n = 21) | | 91 | <LOD | 0.07 | 0.19 | 1.12 | <LOD | 0.10 | 0.17 | 0.58 |
| GAPPS, Yakima | | non-smokers (n = 109) | 0.017 | 93 | <LOD | 0.10 | 0.18 | 44.70 | <LOD | 0.10 | 0.16 | 26.54 |
| | | smokers (n = 19) | | 95 | <LOD | 0.08 | 0.14 | 0.39 | <LOD | 0.09 | 0.11 | 0.28 |
| **2-hydroxyphenanthrene (2-OH-PHEN)** | | | | | | | | | | | | |
| CANDLE, Memphis | | non-smokers (n = 885) | 0.030 | 86 | <LOD | 0.08 | 0.14 | 6.61 | <LOD | 0.08 | 0.13 | 3.72 |
| | | smokers (n = 103) | | 93 | <LOD | 0.13 | 0.22 | 0.91 | <LOD | 0.12 | 0.21 | 0.51 |
| TIDES, San Francisco | | non-smokers (n = 167[a]) | 0.003 | 99 | <LOD | 0.04 | 0.10 | 0.32 | <LOD | 0.06 | 0.10 | 0.39 |
| TIDES, Minneapolis | | non-smokers (n = 163[a]) | 0.003 | 98 | <LOD | 0.05 | 0.10 | 59.00 | <LOD | 0.06 | 0.08 | 101.03 |
| TIDES, Rochester | | non-smokers (n = 145) | 0.003 | 99 | <LOD | 0.09 | 0.17 | 75.00 | <LOD | 0.08 | 0.12 | 293.57 |
| | | smokers (n = 45) | | 98 | <LOD | 0.11 | 0.22 | 0.59 | <LOD | 0.10 | 0.16 | 0.43 |

(*Continued*)

**Table 2.** (Continued)

| Analyte (abbreviation) | Cohort, site | Smoking status | LOD (ng/mL) | Percent detected (%) | OH-PAH concentrations (uncorrected) (ng/mL) | | | | s.g.-corrected OH-PAH concentrations (ng/mL) | | | |
|---|---|---|---|---|---|---|---|---|---|---|---|---|
| | | | | | Min | Median | 75th perc | Max | Min | Median | 75th perc | Max |
| | TIDES, Seattle | non-smokers (n = 135[a]) | 0.003 | 96 | <LOD | 0.04 | 0.07 | 0.58 | <LOD | 0.04 | 0.06 | 0.29 |
| | GAPPS, Seattle | non-smokers (n = 91) | 0.017 | 65 | <LOD | 0.03 | 0.06 | 0.49 | <LOD | 0.03 | 0.05 | 0.48 |
| | | smokers (n = 21) | | 62 | <LOD | 0.03 | 0.07 | 0.43 | <LOD | 0.03 | 0.07 | 0.23 |
| | GAPPS, Yakima | non-smokers (n = 109) | 0.017 | 74 | <LOD | 0.05 | 0.08 | 33.2 | <LOD | 0.04 | 0.07 | 19.70 |
| | | smokers (n = 19) | | 74 | <LOD | 0.05 | 0.07 | 0.17 | <LOD | 0.05 | 0.07 | 0.12 |
| | NHANES 2011–2012[b] | non-smokers (n = 439) | 0.010 | 96 | <LOD | 0.06 | 0.10 | 0.53 | – | – | – | – |
| | | smokers (n = 98) | | 100 | 0.02 | 0.13 | 0.21 | 1.2 | – | – | – | – |
| **3-hydroxyphenanthrene (3-OH-PHEN)** | | | | | | | | | | | | |
| | CANDLE, Memphis | non-smokers (n = 885) | 0.030 | 86 | <LOD | 0.08 | 0.15 | 4.34 | <LOD | 0.09 | 0.14 | 2.69 |
| | | smokers (n = 103) | | 94 | <LOD | 0.18 | 0.30 | 1.62 | <LOD | 0.18 | 0.32 | 1.03 |
| | TIDES, San Francisco | non-smokers (n = 167[a]) | 0.003 | 100 | 0.006 | 0.04 | 0.09 | 0.41 | 0.02 | 0.06 | 0.10 | 0.39 |
| | TIDES, Minneapolis | non-smokers (n = 163[a]) | 0.003 | 96 | <LOD | 0.04 | 0.08 | 0.49 | <LOD | 0.05 | 0.07 | 0.63 |
| | TIDES, Rochester | non-smokers (n = 145) | 0.003 | 98 | <LOD | 0.08 | 0.14 | 0.69 | <LOD | 0.06 | 0.10 | 0.44 |
| | | smokers (n = 45) | | 100 | 0.0031 | 0.16 | 0.28 | 0.72 | 0.01 | 0.14 | 0.23 | 0.73 |
| | TIDES, Seattle | non-smokers (n = 135[a]) | 0.003 | 95 | <LOD | 0.03 | 0.06 | 94.00 | <LOD | 0.04 | 0.05 | 55.99 |
| | GAPPS, Seattle | non-smokers (n = 91) | 0.018 | 65 | <LOD | 0.03 | 0.05 | 0.59 | <LOD | 0.03 | 0.05 | 0.58 |
| | | smokers (n = 21) | | 62 | <LOD | 0.04 | 0.09 | 0.76 | <LOD | 0.04 | 0.07 | 0.39 |
| | GAPPS, Yakima | non-smokers (n = 109) | 0.018 | 77 | <LOD | 0.04 | 0.08 | 20.00 | <LOD | 0.04 | 0.06 | 11.86 |
| | | smokers (n = 19) | | 84 | <LOD | 0.04 | 0.08 | 0.20 | <LOD | 0.04 | 0.06 | 0.14 |
| | NHANES 2011–2012[b] | non-smokers (n = 439) | 0.010 | 94 | <LOD | 0.06 | 0.10 | 0.73 | – | – | – | – |
| | | smokers (n = 98) | | 99 | <LOD | 0.16 | 0.27 | 1.3 | – | – | – | – |
| **4-hydroxyphenanthrene (4-OH-PHEN)** | | | | | | | | | | | | |
| | CANDLE, Memphis | non-smokers (n = 885) | 0.030 | 42 | <LOD | <LOD | 0.05 | 1.26 | <LOD | <LOD | 0.04 | 0.81 |
| | | smokers (n = 103) | | 57 | <LOD | 0.04 | 0.07 | 0.30 | <LOD | 0.03 | 0.07 | 0.30 |
| | TIDES, San Francisco | non-smokers (n = 167[a]) | 0.012 | 72 | <LOD | 0.02 | 0.04 | 0.16 | <LOD | 0.03 | 0.06 | 0.19 |
| | TIDES, Minneapolis | non-smokers (n = 163[a]) | 0.012 | 55 | <LOD | 0.01 | 0.03 | 0.95 | <LOD | 0.02 | 0.03 | 0.76 |
| | TIDES, Rochester | non-smokers (n = 145) | 0.012 | 83 | <LOD | 0.03 | 0.06 | 0.31 | <LOD | 0.03 | 0.04 | 0.23 |
| | | smokers (n = 45) | | 89 | <LOD | 0.04 | 0.08 | 0.34 | <LOD | 0.04 | 0.07 | 0.29 |
| | TIDES, Seattle | non-smokers (n = 135[a]) | 0.012 | 54 | <LOD | 0.01 | 0.02 | 0.17 | <LOD | 0.02 | 0.02 | 0.12 |
| | GAPPS, Seattle | non-smokers (n = 91) | 0.016 | 30 | <LOD | <LOD | 0.02 | 0.24 | <LOD | <LOD | 0.01 | 0.24 |
| | | smokers (n = 21) | | 38 | <LOD | <LOD | 0.02 | 0.16 | <LOD | <LOD | 0.02 | 0.08 |

*(Continued)*

**Table 2.** (Continued)

| Analyte (abbreviation) | Cohort, site | Smoking status | LOD (ng/mL) | Percent detected (%) | OH-PAH concentrations (uncorrected) (ng/mL) | | | | s.g.-corrected OH-PAH concentrations (ng/mL) | | | |
|---|---|---|---|---|---|---|---|---|---|---|---|---|
| | | | | | Min | Median | 75th perc | Max | Min | Median | 75th perc | Max |
| GAPPS, Yakima | | non-smokers (n = 109) | 0.016 | 47 | <LOD | <LOD | 0.03 | 16.64 | <LOD | <LOD | 0.02 | 9.88 |
| | | smokers (n = 19) | | 42 | <LOD | <LOD | 0.03 | 0.06 | <LOD | <LOD | 0.02 | 0.04 |
| NHANES 2011–2012[b] | | non-smokers (n = 438) | 0.010 | 74 | <LOD | 0.02 | 0.03 | 0.56 | – | – | – | – |
| | | smokers (n = 98) | | 92 | <LOD | 0.05 | 0.09 | 0.25 | – | – | – | – |
| **1-hydroxypyrene (1-OH-PYR)** | | | | | | | | | | | | |
| CANDLE, Memphis | | non-smokers (n = 885) | 0.030 | 89 | <LOD | 0.14 | 0.25 | 4.91 | <LOD | 0.14 | 0.23 | 3.04 |
| | | smokers (n = 103) | | 97 | <LOD | 0.25 | 0.51 | 1.44 | <LOD | 0.27 | 0.44 | 1.06 |
| TIDES, San Francisco | | non-smokers (n = 167[a]) | 0.009 | 49 | <LOD | <LOD | 0.34 | 22.00 | <LOD | <LOD | 0.29 | 30.00 |
| TIDES, Minneapolis | | non-smokers (n = 163[a]) | 0.009 | 37 | <LOD | <LOD | 0.10 | 3.09 | <LOD | <LOD | <LOD | 8.45 |
| TIDES, Rochester | | non-smokers (n = 145) | 0.009 | 68 | <LOD | 0.24 | 0.46 | 8.40 | <LOD | 0.18 | 0.34 | 6.39 |
| | | smokers (n = 45) | | 73 | <LOD | 0.43 | 0.85 | 3.69 | <LOD | 0.35 | 0.61 | 2.42 |
| TIDES, Seattle | | non-smokers (n = 135[a]) | 0.009 | 22 | <LOD | <LOD | <LOD | 3.92 | <LOD | <LOD | <LOD | 2.07 |
| GAPPS, Seattle | | non-smokers (n = 91) | 0.020 | 56 | <LOD | 0.03 | 0.08 | 0.42 | <LOD | 0.04 | 0.07 | 0.80 |
| | | smokers (n = 21) | | 62 | <LOD | 0.03 | 0.14 | 0.42 | <LOD | 0.05 | 0.13 | 0.34 |
| GAPPS, Yakima | | non-smokers (n = 109) | 0.020 | 47 | <LOD | <LOD | 0.08 | 14.22 | <LOD | <LOD | 0.07 | 8.45 |
| | | smokers (n = 19) | | 74 | <LOD | 0.06 | 0.11 | 0.75 | <LOD | 0.06 | 0.09 | 0.61 |
| NHANES 2011–2012[b] | | non-smokers (n = 438) | 0.010 | 99 | <LOD | 0.11 | 0.22 | 2.6 | – | – | – | – |
| | | smokers (n = 98) | | 99 | <LOD | 0.31 | 0.50 | 2.5 | – | – | – | – |
| **3-hydroxybenzo(c)phenanthrene (3-OH-BCP)** | | | | | | | | | | | | |
| CANDLE, Memphis | | non-smokers (n = 885) | 0.025 | 0 | – | – | – | – | – | – | – | – |
| | | smokers (n = 103) | | <1 | – | – | – | – | – | – | – | – |
| TIDES, San Francisco | | non-smokers (n = 167[a]) | 0.005 | <1 | – | – | – | – | – | – | – | – |
| TIDES, Minneapolis | | non-smokers (n = 163[a]) | 0.005 | 0 | – | – | – | – | – | – | – | – |
| TIDES, Rochester | | non-smokers (n = 145) | 0.005 | 0 | – | – | – | – | – | – | – | – |
| | | smokers (n = 45) | | 0 | – | – | – | – | – | – | – | – |
| TIDES, Seattle | | non-smokers (n = 135[a]) | 0.005 | 0 | – | – | – | – | – | – | – | – |
| GAPPS, Seattle | | non-smokers (n = 91) | 0.020 | 0 | – | – | – | – | – | – | – | – |
| | | smokers (n = 21) | | 0 | – | – | – | – | – | – | – | – |
| GAPPS, Yakima | | non-smokers (n = 109) | 0.020 | 0 | – | – | – | – | – | – | – | – |
| | | smokers (n = 19) | | 0 | – | – | – | – | – | – | – | – |
| **1-hydroxychrysene (1-OH-CHRY)** | | | | | | | | | | | | |

*(Continued)*

**Table 2.** (*Continued*)

| Analyte (abbreviation) | Cohort, site | Smoking status | LOD (ng/mL) | Percent detected (%) | OH-PAH concentrations (uncorrected) (ng/mL) | | | | s.g.-corrected OH-PAH concentrations (ng/mL) | | | |
|---|---|---|---|---|---|---|---|---|---|---|---|---|
| | | | | | Min | Median | 75th perc | Max | Min | Median | 75th perc | Max |
| CANDLE, Memphis | | non-smokers (n = 885) | 0.020 | 0 | – | – | – | – | – | – | – | – |
| | | smokers (n = 103) | | 0 | – | – | – | – | – | – | – | – |
| TIDES, San Francisco | | non-smokers (n = 167[a]) | 0.072 | 1 | – | – | – | – | – | – | – | – |
| TIDES, Minneapolis | | non-smokers (n = 163[a]) | 0.072 | 0 | – | – | – | – | – | – | – | – |
| TIDES, Rochester | | non-smokers (n = 145) | 0.072 | <1 | – | – | – | – | – | – | – | – |
| | | smokers (n = 45) | | 0 | – | – | – | – | – | – | – | – |
| TIDES, Seattle | | non-smokers (n = 135[a]) | 0.072 | 2 | – | – | – | – | – | – | – | – |
| GAPPS, Seattle | | non-smokers (n = 91) | 0.021 | 0 | – | – | – | – | – | – | – | – |
| | | smokers (n = 21) | | 0 | – | – | – | – | – | – | – | – |
| GAPPS, Yakima | | non-smokers (n = 109) | 0.021 | 0 | – | – | – | – | – | – | – | – |
| | | smokers (n = 19) | | 0 | – | – | – | – | – | – | – | – |
| **6-hydroxychrysene (6-OH-CHRY)** | | | | | | | | | | | | |
| CANDLE, Memphis | | non-smokers (n = 885) | 0.025 | 0 | – | – | – | – | – | – | – | – |
| | | smokers (n = 103) | | 0 | – | – | – | – | – | – | – | – |
| TIDES, San Francisco | | non-smokers (n = 167[a]) | 0.011 | 6 | – | – | – | – | – | – | – | – |
| TIDES, Minneapolis | | non-smokers (n = 163[a]) | 0.011 | <1 | – | – | – | – | – | – | – | – |
| TIDES, Rochester | | non-smokers (n = 145) | 0.011 | 1 | – | – | – | – | – | – | – | – |
| | | smokers (n = 45) | | 0 | – | – | – | – | – | – | – | – |
| TIDES, Seattle | | non-smokers (n = 135[a]) | 0.011 | 0 | – | – | – | – | – | – | – | – |
| GAPPS, Seattle | | non-smokers (n = 91) | 0.019 | 0 | – | – | – | – | – | – | – | – |
| | | smokers (n = 21) | | 0 | – | – | – | – | – | – | – | – |
| GAPPS, Yakima | | non-smokers (n = 109) | 0.019 | 0 | – | – | – | – | – | – | – | – |
| | | smokers (n = 19) | | 0 | – | – | – | – | – | – | – | – |
| **1-hydroxybenz(a)anthracene (1-OH-BAA)** | | | | | | | | | | | | |
| CANDLE, Memphis | | non-smokers (n = 885) | 0.030 | 0 | – | – | – | – | – | – | – | – |
| | | smokers (n = 103) | | 0 | – | – | – | – | – | – | – | – |
| TIDES, San Francisco | | non-smokers (n = 167[a]) | 0.016 | 2 | – | – | – | – | – | – | – | – |
| TIDES, Minneapolis | | non-smokers (n = 163[a]) | 0.016 | 1 | – | – | – | – | – | – | – | – |
| TIDES, Rochester | | non-smokers (n = 145) | 0.016 | 0 | – | – | – | – | – | – | – | – |
| | | smokers (n = 45) | | 0 | – | – | – | – | – | – | – | – |

(*Continued*)

**Table 2.** (Continued)

| Analyte (abbreviation) | Cohort, site | Smoking status | LOD (ng/mL) | Percent detected (%) | OH-PAH concentrations (uncorrected) (ng/mL) | | | | s.g.-corrected OH-PAH concentrations (ng/mL) | | | |
|---|---|---|---|---|---|---|---|---|---|---|---|---|
| | | | | | Min | Median | 75th perc | Max | Min | Median | 75th perc | Max |
| | TIDES, Seattle | non-smokers (n = 135[a]) | 0.016 | 0 | – | – | – | – | – | – | – | – |
| | GAPPS, Seattle | non-smokers (n = 91) | 0.019 | 0 | – | – | – | – | – | – | – | – |
| | | smokers (n = 21) | | 0 | – | – | – | – | – | – | – | – |
| | GAPPS, Yakima | non-smokers (n = 109) | 0.019 | 0 | – | – | – | – | – | – | – | – |
| | | smokers (n = 19) | | 0 | – | – | – | – | – | – | – | – |

[a]We do not report data for smokers at the following TIDES sites due to small sub-sample sizes: San Francisco (n = 2), Minneapolis (n = 4), Seattle (n = 3).

[b]NHANES 2011–2012, females aged 16–49 years; not weighted to be representative of the U.S. civilian noninstitutionalized resident population.

were low at all sites except CANDLE, where it was found in 88.8% and 97.1% of non-smoking and smoking during pregnancy samples, respectively. 3-hydroxybenzo(c)phenanthrene, 1- and 6-hydroxychrysene, and 1-hydroxybenz(a)anthracene were either not detected or detected in only a few samples and are not discussed further.

## Site comparisons

Fig 1 shows the s.g.-corrected GM OH-PAH concentrations by site (exact values are reported in Table 3). Among the 8 metabolites we analyzed, 2-hydroxynaphthalene was detected in the highest concentrations at all sites, with GMs ranging from 1.73–4.95 ng/mL across sites. At

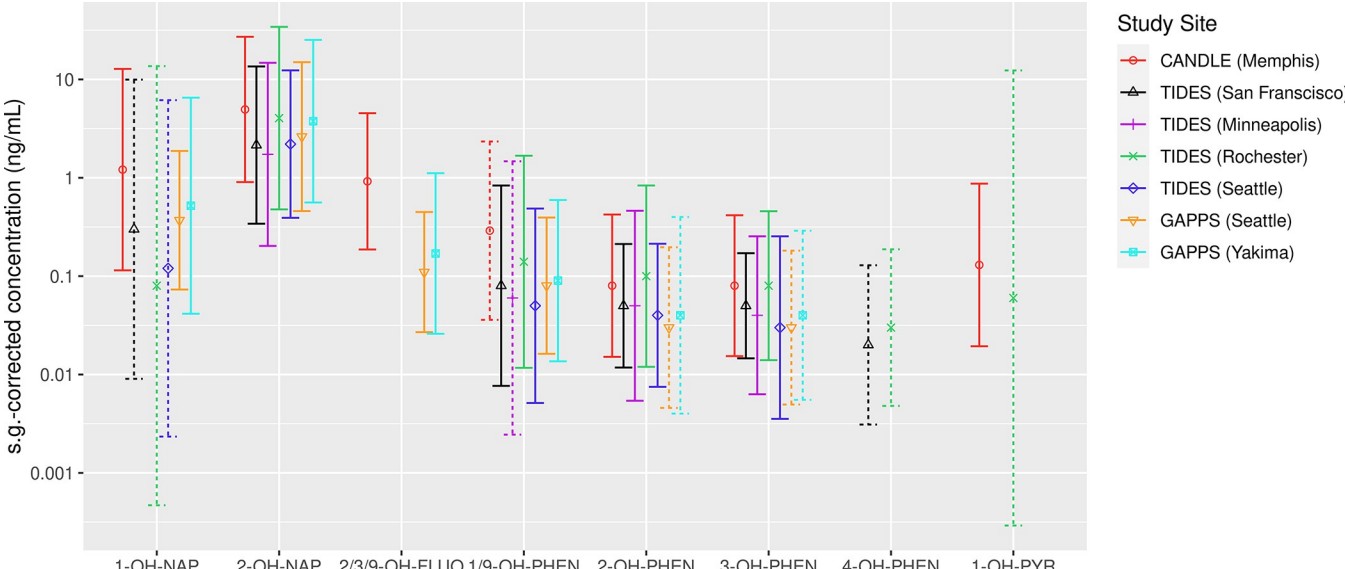

**Fig 1. Geometric mean (95% confidence interval) of s.g.-corrected concentrations (ng/mL) of 8 OH-PAHs from mid pregnancy visit among non-smoking individuals, by study site (n = 1,695).** Analytes with detection frequencies <60% are not shown; dashed lines indicate 60–85% detection frequencies, interpret with caution; solid lines indicate >85% detection frequencies, very confident. s.g., specific gravity. Analyte abbreviations: 1-OH-NAP, 1-hydroxynaphthalene; 2-OH-NAP, 2-hydroxynaphthalene; 2/3/9-OH-FLUO, 2/3/9-hydroxyfluorene; 1/9-OH-PHEN, 1/9-hydroxyphenanthrene; 2-OH-PHEN, 2-hydroxyphenanthrene; 3-OH-PHEN, 3-hydroxyphenanthrene; 4-OH-PHEN, 4-hydroxyphenanthrene; 1-OH-PYR, 1-hydroxypyrene.

**Table 3. Geometric mean (GSD) of s.g.-corrected concentrations (ng/mL) of 8 OH-PAHs from mid pregnancy visit among non-smoking individuals, by socioeconomic characteristics and study site (n = 1,695) (analytes detected in < 60% of samples not shown).**

| Socioeconomic characteristic | CANDLE (Memphis) | | TIDES (San Francisco) | | TIDES (Minneapolis) | | TIDES (Rochester) | | TIDES (Seattle) | | GAPPS (Seattle) | | GAPPS (Yakima) | |
|---|---|---|---|---|---|---|---|---|---|---|---|---|---|---|
| | GM (ng/mL) | GSD | GM (ng/mL) | GSD | GM (ng/mL) | GSD | GM (ng/mL) | GSD | GM (ng/mL) | GSD | GM (ng/mL) | GSD | GM (ng/mL) | GSD |
| **1-hydroxynaphthalene (1-OH-NAP)** | | | | | | | | | | | | | | |
| Overall | 1.21 | 3.25 | 0.30 | 5.76 | – | – | 0.08 | 13.1 | 0.12 | 7.16 | 0.37 | 2.25 | 0.52 | 3.54 |
| Income (USD, k = 1,000) | | | | | | | | | | | | | | |
| < 15k (< 20k GAPPS) | 1.52 | 3.94 | a | a | – | – | 0.07 | 16.8 | 0.06 | 15.6 | 0.29 | 1.98 | 0.57 | 4.18 |
| 15k–25k (20k–30k GAPPS) | 1.32 | 3.24 | a | a | – | – | 0.10 | 12.8 | a | a | a | a | 0.50 | 10.24 |
| 25k–45k (30k–50k GAPPS) | 1.20 | 3.37 | 0.55 | 5.08 | – | – | 0.06 | 12.0 | 0.04 | 7.01 | 0.37 | 2.61 | 0.69 | 3.27 |
| 45k–55k (50k–60k GAPPS) | 1.25 | 2.80 | a | a | – | – | a | a | 0.01 | 21.4 | a | a | 1.00 | 3.19 |
| 55k–65k (60k–70k GAPPS) | 0.91 | 3.11 | 0.19 | 14.37 | – | – | 0.14 | 3.01 | 0.24 | 3.69 | a | a | 0.45 | 1.98 |
| 65k–75k (70k–80k GAPPS) | 1.22 | 2.82 | 0.18 | 2.63 | – | – | 0.27 | 6.13 | 0.07 | 7.10 | 0.16 | 1.92 | a | a |
| ≥ 75k (80k GAPPS) | 0.84 | 2.29 | 0.32 | 5.33 | – | – | 0.07 | 9.70 | 0.21 | 5.49 | 0.32 | 2.14 | 0.48 | 2.62 |
| Education | | | | | | | | | | | | | | |
| < High school | 1.64 | 3.87 | a | a | – | – | 0.04 | 12.8 | a | a | a | a | a | a |
| High school completion | 1.35 | 3.40 | 0.07 | 13.9 | – | – | 0.13 | 14.4 | 0.05 | 8.53 | 0.19 | 2.27 | 0.60 | 2.99 |
| Graduated college/technical school | 3.40 | 3.02 | 0.44 | 4.85 | – | – | 0.05 | 14.4 | 0.14 | 5.11 | 0.31 | 2.10 | 0.50 | 4.40 |
| Some graduate work or graduate/professional degree | 0.93 | 2.55 | 0.27 | 5.65 | – | – | 0.09 | 7.82 | 0.13 | 8.38 | 0.31 | 2.45 | 0.61 | 2.40 |
| Race | | | | | | | | | | | | | | |
| White | 0.94 | 2.39 | 0.44 | 4.90 | – | – | 0.06 | 9.38 | 0.17 | 5.99 | 0.40 | 2.25 | 0.52 | 2.98 |
| Black/African American | 1.41 | 3.61 | a | a | – | – | 0.07 | 18.52 | a | a | a | a | a | a |
| Asian/Other/Multiple | 0.41 | 4.88 | 0.30 | 6.49 | – | – | 0.30 | 6.49 | 0.03 | 28.96 | 0.27 | 2.13 | 0.64 | 3.49 |
| Ethnicity | | | | | | | | | | | | | | |
| Not Hispanic or Latino | 1.24 | 3.26 | 0.36 | 6.58 | – | – | 0.08 | 12.4 | 0.15 | 7.33 | 0.36 | 2.12 | 0.53 | 2.89 |
| Hispanic/Latino | 0.97 | 3.88 | 0.24 | 7.34 | – | – | 0.01 | 36.28 | 0.02 | 16.19 | 0.45 | 3.23 | 0.61 | 4.58 |
| **2-hydroxynaphthalene (2-OH-NAP)** | | | | | | | | | | | | | | |
| Overall | 4.95 | 2.34 | 2.15 | 2.51 | 1.73 | 2.92 | 4.04 | 2.91 | 2.20 | 2.37 | 2.62 | 2.39 | 3.76 | 2.59 |
| Income (USD, k = 1,000) | | | | | | | | | | | | | | |
| < 15k (< 20k GAPPS) | 5.95 | 2.70 | a | a | 2.52 | 3.22 | 5.15 | 2.87 | 2.90 | 2.41 | 3.33 | 1.90 | 3.71 | 1.80 |
| 15k–25k (20k–30k GAPPS) | 5.67 | 2.05 | a | a | 2.09 | 4.82 | 4.46 | 1.86 | a | a | a | a | 5.67 | 4.19 |
| 25k–45k (30k–50k GAPPS) | 4.71 | 2.16 | 5.33 | 3.21 | 2.71 | 1.83 | 4.05 | 2.67 | 1.78 | 2.05 | 2.81 | 3.53 | 5.89 | 1.94 |
| 45k–55k (50k–60k GAPPS) | 5.28 | 2.41 | a | a | 1.50 | 3.07 | a | a | 4.46 | 2.20 | a | a | 4.64 | 1.97 |
| 55k–65k (60k–70k GAPPS) | 4.07 | 2.28 | 2.95 | 3.47 | 1.54 | 5.35 | 2.52 | 1.95 | 1.70 | 2.09 | a | a | 3.51 | 3.32 |
| 65k–75k (70k–80k GAPPS) | 4.08 | 2.11 | 1.31 | 1.70 | 1.23 | 2.75 | 0.86 | 9.59 | 2.50 | 2.33 | 1.32 | 1.94 | a | a |
| ≥ 75k (80k GAPPS) | 3.65 | 2.13 | 2.10 | 2.44 | 1.60 | 2.37 | 3.13 | 2.58 | 1.99 | 2.27 | 1.95 | 2.28 | 3.17 | 2.26 |
| Education | | | | | | | | | | | | | | |
| < High school | 7.17 | 2.27 | a | a | a | a | 4.93 | 2.16 | a | a | a | a | a | a |
| High school completion | 5.66 | 2.22 | 3.75 | 2.64 | 2.05 | 4.27 | 4.83 | 2.70 | 4.00 | 2.24 | 2.02 | 2.01 | 4.30 | 2.67 |
| Graduated college/technical school | 4.42 | 2.26 | 2.12 | 2.67 | 2.19 | 2.44 | 4.27 | 2.86 | 2.18 | 2.29 | 2.43 | 2.57 | 4.08 | 2.51 |
| Some graduate work or graduate/professional degree | 3.32 | 2.16 | 2.02 | 2.34 | 1.33 | 2.72 | 2.25 | 3.52 | 1.94 | 2.32 | 1.54 | 1.99 | 3.05 | 2.10 |
| Race | | | | | | | | | | | | | | |
| White | 3.60 | 2.15 | 2.56 | 2.52 | 1.94 | 2.76 | 2.06 | 3.12 | 2.20 | 2.41 | 2.58 | 2.32 | 3.49 | 2.62 |
| Black/African American | 6.11 | 2.27 | a | a | 2.40 | 5.69 | 4.66 | 2.39 | a | a | a | a | a | a |
| Asian/Other/Multiple | 1.18 | 3.29 | 3.14 | 2.52 | 1.75 | 1.92 | 3.14 | 2.52 | 2.92 | 2.05 | 2.31 | 2.40 | 4.85 | 2.07 |
| Ethnicity | | | | | | | | | | | | | | |

*(Continued)*

**Table 3.** (Continued)

| Socioeconomic characteristic | CANDLE (Memphis) | | TIDES (San Francisco) | | TIDES (Minneapolis) | | TIDES (Rochester) | | TIDES (Seattle) | | GAPPS (Seattle) | | GAPPS (Yakima) | |
|---|---|---|---|---|---|---|---|---|---|---|---|---|---|---|
| | GM (ng/mL) | GSD | GM (ng/mL) | GSD | GM (ng/mL) | GSD | GM (ng/mL) | GSD | GM (ng/mL) | GSD | GM (ng/mL) | GSD | GM (ng/mL) | GSD |
| Not Hispanic or Latino | 5.06 | 2.33 | 0.43 | 4.88 | 1.88 | 2.77 | 2.82 | 2.91 | 2.26 | 2.35 | 2.55 | 2.38 | 3.54 | 2.64 |
| Hispanic/Latino | 3.43 | 5.89 | 0.31 | 4.91 | a | a | 5.13 | 2.47 | 3.07 | 2.60 | 3.09 | 2.70 | 4.79 | 2.36 |
| **2-hydroxyfluorene/3-hydroxyfluorene/9-hydroxyfluorene (2/3/9-OH-FLUO)** | | | | | | | | | | | | | | |
| Overall | 0.92 | 2.22 | – | – | – | – | – | – | – | – | 0.11 | 2.02 | 0.17 | 2.56 |
| Income (USD, k = 1,000) | | | | | | | | | | | | | | |
| < 15k (< 20k GAPPS) | 1.10 | 2.15 | – | – | – | – | – | – | – | – | 0.12 | 1.79 | 0.20 | 2.40 |
| 15k–25k (20k–30k GAPPS) | 1.08 | 1.90 | – | – | – | – | – | – | – | – | a | a | 0.25 | 2.52 |
| 25k–45k (30k–50k GAPPS) | 0.95 | 1.99 | – | – | – | – | – | – | – | – | 0.10 | 1.88 | 0.22 | 1.44 |
| 45k–55k (50k–60k GAPPS) | 0.96 | 2.39 | – | – | – | – | – | – | – | – | a | a | 0.27 | 5.60 |
| 55k–65k (60k–70k GAPPS) | 0.80 | 1.90 | – | – | – | – | – | – | – | – | a | a | 0.16 | 2.44 |
| 65k–75k (70k–80k GAPPS) | 0.74 | 2.73 | – | – | – | – | – | – | – | – | 0.03 | 4.18 | a | a |
| ≥ 75k (80k GAPPS) | 0.61 | 2.49 | – | – | – | – | – | – | – | – | 0.09 | 1.76 | 0.14 | 2.17 |
| Education | | | | | | | | | | | | | | |
| < High school | 1.21 | 2.15 | – | – | – | – | – | – | – | – | a | a | a | a |
| High school completion | 1.05 | 1.97 | – | – | – | – | – | – | – | – | 0.08 | 2.12 | 0.18 | 2.20 |
| Graduated college/technical school | 0.86 | 2.17 | – | – | – | – | – | – | – | – | 0.09 | 2.06 | 0.19 | 3.25 |
| Some graduate work or graduate/professional degree | 0.55 | 2.79 | – | – | – | – | – | – | – | – | 0.10 | 1.74 | 0.16 | 1.57 |
| Race | | | | | | | | | | | | | | |
| White | 0.70 | 2.14 | – | – | – | – | – | – | – | – | 0.12 | 1.74 | 0.17 | 2.39 |
| Black/African American | 1.15 | 2.04 | – | – | – | – | – | – | – | – | a | a | a | a |
| Asian/Other/Multiple | 0.65 | 2.28 | – | – | – | – | – | – | – | – | 0.10 | 2.10 | 0.21 | 2.33 |
| Ethnicity | | | | | | | | | | | | | | |
| Not Hispanic or Latino | 0.96 | 2.15 | – | – | – | – | – | – | – | – | 0.12 | 1.79 | 0.16 | 1.88 |
| Hispanic/Latino | 0.76 | 2.93 | – | – | – | – | – | – | – | – | 0.12 | 1.99 | 0.24 | 4.15 |
| **1-hydroxyphenanthrene/9-hydroxyphenanthrene (1/9-OH-PHEN)** | | | | | | | | | | | | | | |
| Overall | 0.29 | 2.84 | 0.08 | 3.23 | 0.06 | 4.95 | 0.14 | 3.46 | 0.05 | 3.12 | 0.08 | 2.22 | 0.09 | 2.57 |
| Income (USD, k = 1,000) | | | | | | | | | | | | | | |
| < 15k (< 20k GAPPS) | 0.29 | 2.74 | a | a | 0.10 | 3.19 | 0.17 | 2.61 | 0.09 | 2.12 | 0.09 | 2.69 | 0.11 | 1.85 |
| 15k–25k (20k–30k GAPPS) | 0.33 | 2.42 | a | a | 0.03 | 4.87 | 0.12 | 5.36 | a | a | a | a | 0.10 | 2.52 |
| 25k–45k (30k–50k GAPPS) | 0.27 | 3.09 | 0.10 | 4.40 | 0.05 | 4.34 | 0.16 | 2.29 | 0.09 | 1.98 | 0.07 | 1.92 | 0.11 | 2.34 |
| 45k–55k (50k–60k GAPPS) | 0.26 | 3.01 | a | a | 0.04 | 3.81 | a | a | 0.05 | 2.06 | a | a | 0.21 | 5.11 |
| 55k–65k (60k–70k GAPPS) | 0.30 | 2.48 | 0.13 | 1.91 | 0.05 | 3.22 | 0.20 | 1.63 | 0.05 | 3.79 | a | a | 0.09 | 2.29 |
| 65k–75k (70k–80k GAPPS) | 0.31 | 3.43 | 0.06 | 3.10 | 0.02 | 8.65 | 0.08 | 4.90 | 0.04 | 4.08 | 0.05 | 2.27 | a | a |
| ≥ 75k (80k GAPPS) | 0.23 | 3.02 | 0.08 | 3.26 | 0.07 | 5.03 | 0.13 | 4.08 | 0.05 | 3.23 | 0.06 | 2.11 | 0.08 | 2.02 |
| Education | | | | | | | | | | | | | | |
| < High school | 0.31 | 3.21 | a | a | a | a | 0.15 | 2.90 | a | a | a | a | a | a |
| High school completion | 0.32 | 2.47 | 0.04 | 5.03 | 0.02 | 6.78 | 0.17 | 2.84 | 0.06 | 3.37 | 0.06 | 1.62 | 0.10 | 2.17 |
| Graduated college/technical school | 0.27 | 3.16 | 0.07 | 3.64 | 0.07 | 4.57 | 0.09 | 4.36 | 0.04 | 3.34 | 0.05 | 2.42 | 0.10 | 3.42 |
| Some graduate work or graduate/professional degree | 0.22 | 3.27 | 0.09 | 2.85 | 0.05 | 5.08 | 0.15 | 4.23 | 0.05 | 2.92 | 0.07 | 1.98 | 0.09 | 1.65 |
| Race | | | | | | | | | | | | | | |
| White | 0.23 | 3.69 | 0.11 | 2.83 | 0.07 | 4.95 | 0.10 | 3.35 | 0.06 | 2.65 | 0.09 | 2.09 | 0.10 | 2.35 |
| Black/African American | 0.26 | 3.95 | a | a | 0.07 | 2.50 | 0.13 | 3.01 | a | a | a | a | a | a |
| Asian/Other/Multiple | 0.27 | 3.48 | 0.07 | 3.84 | 0.06 | 2.86 | 0.07 | 3.84 | 0.04 | 2.89 | 0.06 | 1.64 | 0.12 | 1.66 |

(*Continued*)

**Table 3.** (*Continued*)

| Socioeconomic characteristic | CANDLE (Memphis) | | TIDES (San Francisco) | | TIDES (Minneapolis) | | TIDES (Rochester) | | TIDES (Seattle) | | GAPPS (Seattle) | | GAPPS (Yakima) | |
|---|---|---|---|---|---|---|---|---|---|---|---|---|---|---|
| | GM (ng/mL) | GSD | GM (ng/mL) | GSD | GM (ng/mL) | GSD | GM (ng/mL) | GSD | GM (ng/mL) | GSD | GM (ng/mL) | GSD | GM (ng/mL) | GSD |
| Ethnicity | | | | | | | | | | | | | | |
| Not Hispanic or Latino | 0.25 | 3.87 | 0.10 | 3.25 | 0.07 | 4.64 | 0.11 | 3.19 | 0.06 | 2.66 | 0.08 | 1.97 | 0.10 | 1.72 |
| Hispanic/Latino | 0.34 | 2.99 | 0.12 | 1.77 | a | a | 0.12 | 2.38 | 0.05 | 3.79 | 0.07 | 2.28 | 0.12 | 4.16 |
| **2-hydroxyphen[a]nthrene (2-OH-PHEN)** | | | | | | | | | | | | | | |
| Overall | 0.08 | 2.30 | 0.05 | 2.06 | 0.05 | 3.04 | 0.10 | 2.89 | 0.04 | 2.31 | 0.03 | 2.56 | 0.04 | 3.16 |
| Income (USD, k = 1,000) | | | | | | | | | | | | | | |
| < 15k (< 20k GAPPS) | 0.09 | 2.04 | a | a | 0.06 | 1.93 | 0.13 | 3.94 | 0.06 | 1.68 | 0.04 | 1.92 | 0.04 | 2.33 |
| 15k–25k (20k–30k GAPPS) | 0.09 | 2.10 | a | a | 0.04 | 2.25 | 0.11 | 2.69 | a | a | a | a | 0.05 | 1.99 |
| 25k–45k (30k–50k GAPPS) | 0.09 | 2.23 | 0.09 | 1.51 | 0.05 | 1.51 | 0.11 | 1.78 | 0.05 | 1.90 | 0.02 | 2.73 | 0.05 | 2.27 |
| 45k–55k (50k–60k GAPPS) | 0.08 | 2.50 | a | a | 0.07 | 7.33 | a | a | 0.04 | 1.99 | a | a | 0.10 | 5.69 |
| 55k–65k (60k–70k GAPPS) | 0.08 | 2.13 | 0.05 | 2.02 | 0.03 | 2.57 | 0.09 | 1.52 | 0.04 | 1.77 | a | a | 0.03 | 3.40 |
| 65k–75k (70k–80k GAPPS) | 0.07 | 2.80 | 0.04 | 1.44 | 0.03 | 2.38 | 0.07 | 1.74 | 0.04 | 2.75 | 0.02 | 1.88 | a | a |
| ≥ 75k (80k GAPPS) | 0.05 | 2.53 | 0.05 | 2.10 | 0.04 | 2.89 | 0.08 | 2.10 | 0.03 | 2.44 | 0.02 | 2.27 | 0.03 | 2.63 |
| Education | | | | | | | | | | | | | | |
| < High school | 0.08 | 2.18 | a | a | a | a | 0.13 | 5.12 | a | a | a | a | a | a |
| High school completion | 0.09 | 1.98 | 0.04 | 2.65 | 0.03 | 2.35 | 0.11 | 2.59 | 0.06 | 1.87 | 0.02 | 1.91 | 0.04 | 2.35 |
| Graduated college/technical school | 0.08 | 2.52 | 0.05 | 2.01 | 0.05 | 2.48 | 0.09 | 2.21 | 0.03 | 2.74 | 0.02 | 2.49 | 0.04 | 4.44 |
| Some graduate work or graduate/professional degree | 0.05 | 2.97 | 0.05 | 1.99 | 0.04 | 3.63 | 0.09 | 1.83 | 0.04 | 1.99 | 0.03 | 2.05 | 0.03 | 2.20 |
| Race | | | | | | | | | | | | | | |
| White | 0.07 | 2.00 | 0.06 | 2.08 | 0.06 | 3.08 | 0.07 | 2.14 | 0.04 | 2.11 | 0.03 | 2.18 | 0.04 | 3.05 |
| Black/African American | 0.10 | 2.05 | a | a | 0.04 | 2.21 | 0.11 | 3.74 | a | a | a | a | a | a |
| Asian/Other/Multiple | 0.07 | 2.05 | 0.06 | 1.77 | 0.04 | 1.85 | 0.07 | 1.90 | 0.03 | 2.46 | 0.03 | 2.20 | 0.06 | 1.71 |
| Ethnicity | | | | | | | | | | | | | | |
| Not Hispanic or Latino | 0.09 | 2.06 | 0.06 | 2.03 | 0.05 | 2.95 | 0.08 | 2.90 | 0.04 | 2.18 | 0.03 | 2.22 | 0.04 | 2.15 |
| Hispanic/Latino | 0.08 | 2.18 | 0.07 | 2.05 | a | a | 0.08 | 2.07 | 0.05 | 1.94 | 0.03 | 2.03 | 0.06 | 4.61 |
| **3-hydroxyphenanthrene (3-OH-PHEN)** | | | | | | | | | | | | | | |
| Overall | 0.08 | 2.28 | 0.05 | 1.85 | 0.04 | 2.52 | 0.08 | 2.39 | 0.03 | 2.91 | 0.03 | 2.46 | 0.04 | 2.69 |
| Income (USD, k = 1,000) | | | | | | | | | | | | | | |
| < 15k (< 20k GAPPS) | 0.10 | 1.95 | a | a | 0.05 | 1.90 | 0.10 | 2.11 | 0.05 | 2.23 | 0.04 | 1.92 | 0.04 | 2.33 |
| 15k–25k (20k–30k GAPPS) | 0.09 | 2.02 | a | a | 0.03 | 1.69 | 0.10 | 2.24 | a | a | a | a | 0.05 | 2.03 |
| 25k–45k (30k–50k GAPPS) | 0.10 | 2.24 | 0.08 | 1.31 | 0.05 | 1.56 | 0.09 | 2.04 | 0.04 | 1.70 | 0.02 | 2.40 | 0.05 | 1.97 |
| 45k–55k (50k–60k GAPPS) | 0.08 | 2.41 | a | a | 0.03 | 2.86 | a | a | 0.03 | 2.01 | a | a | 0.08 | 5.34 |
| 55k–65k (60k–70k GAPPS) | 0.07 | 2.00 | 0.05 | 1.96 | 0.03 | 1.59 | 0.07 | 1.50 | 0.02 | 3.24 | a | a | 0.03 | 2.72 |
| 65k–75k (70k–80k GAPPS) | 0.05 | 2.85 | 0.05 | 1.36 | 0.02 | 3.30 | 0.04 | 4.54 | 0.03 | 2.56 | 0.02 | 2.18 | a | a |
| ≥ 75k (80k GAPPS) | 2.85 | 2.65 | 0.05 | 1.87 | 0.04 | 2.75 | 0.06 | 2.99 | 0.03 | 3.36 | 0.02 | 2.25 | 0.03 | 2.20 |
| Education | | | | | | | | | | | | | | |
| < High school | 0.10 | 2.15 | a | a | a | a | 0.10 | 1.97 | a | a | a | a | a | a |
| High school completion | 0.10 | 1.92 | 0.04 | 1.78 | 0.02 | 3.34 | 0.09 | 2.14 | 0.04 | 2.07 | 0.02 | 1.85 | 0.04 | 2.14 |
| Graduated college/technical school | 0.08 | 2.49 | 0.05 | 1.85 | 0.05 | 2.20 | 0.06 | 3.17 | 0.04 | 3.59 | 0.02 | 2.39 | 0.04 | 3.74 |
| Some graduate work or graduate/professional degree | 0.04 | 3.03 | 0.05 | 1.84 | 0.04 | 2.59 | 0.07 | 2.23 | 0.03 | 2.43 | 0.03 | 2.02 | 0.04 | 1.75 |
| Race | | | | | | | | | | | | | | |
| White | 0.07 | 2.01 | 0.06 | 1.89 | 0.05 | 2.57 | 0.05 | 2.21 | 0.04 | 2.80 | 0.03 | 2.19 | 0.04 | 2.51 |
| Black/African American | 0.11 | 1.98 | a | a | 0.03 | 1.89 | 0.11 | 3.79 | a | a | a | a | a | a |

(*Continued*)

**Table 3.** (*Continued*)

| Socioeconomic characteristic | CANDLE (Memphis) | | TIDES (San Francisco) | | TIDES (Minneapolis) | | TIDES (Rochester) | | TIDES (Seattle) | | GAPPS (Seattle) | | GAPPS (Yakima) | |
|---|---|---|---|---|---|---|---|---|---|---|---|---|---|---|
| | GM (ng/mL) | GSD | GM (ng/mL) | GSD | GM (ng/mL) | GSD | GM (ng/mL) | GSD | GM (ng/mL) | GSD | GM (ng/mL) | GSD | GM (ng/mL) | GSD |
| Asian/Other/Multiple | 0.07 | 2.15 | 0.06 | 1.77 | 0.04 | 1.88 | 0.06 | 1.77 | 0.03 | 2.02 | 0.03 | 1.80 | 0.05 | 1.71 |
| Ethnicity | | | | | | | | | | | | | | |
| Not Hispanic or Latino | 0.09 | 2.04 | 0.06 | 1.86 | 0.05 | 2.51 | 0.06 | 2.24 | 0.04 | 2.76 | 0.03 | 2.14 | 0.04 | 1.78 |
| Hispanic/Latino | 0.07 | 2.25 | 0.07 | 1.82 | a | a | 0.06 | 1.82 | 0.04 | 1.97 | 0.03 | 1.80 | 0.05 | 4.19 |
| **4-hydroxyphenanthrene (4-OH-PHEN)** | | | | | | | | | | | | | | |
| Overall | – | – | 0.02 | 2.54 | – | – | 0.03 | 2.50 | – | – | – | – | – | – |
| Income (USD, k = 1,000) | | | | | | | | | | | | | | |
| < 15k (< 20k GAPPS) | – | – | a | a | – | – | 0.04 | 2.11 | – | – | – | – | – | – |
| 15k–25k (20k–30k GAPPS) | – | – | a | a | – | – | 0.03 | 3.39 | – | – | – | – | – | – |
| 25k–45k (30k–50k GAPPS) | – | – | 0.04 | 2.23 | – | – | 0.03 | 2.01 | – | – | – | – | – | – |
| 45k–55k (50k–60k GAPPS) | – | – | a | a | – | – | a | a | – | – | – | – | – | – |
| 55k–65k (60k–70k GAPPS) | – | – | 0.02 | 3.05 | – | – | 0.03 | 1.80 | – | – | – | – | – | – |
| 65k–75k (70k–80k GAPPS) | – | – | 0.02 | 2.03 | – | – | 0.02 | 2.82 | – | – | – | – | – | – |
| ≥ 75k (80k GAPPS) | – | – | 0.02 | 2.54 | – | – | 0.02 | 2.75 | – | – | – | – | – | – |
| Education | | | | | | | | | | | | | | |
| < High school | – | – | a | a | – | – | 0.04 | 2.26 | – | – | – | – | – | – |
| High school completion | – | – | 0.02 | 2.54 | – | – | 0.04 | 2.41 | – | – | – | – | – | – |
| Graduated college/technical school | – | – | 0.02 | 2.65 | – | – | 0.03 | 2.92 | – | – | – | – | – | – |
| Some graduate work or graduate/professional degree | – | – | 0.02 | 2.51 | – | – | 0.02 | 2.14 | – | – | – | – | – | – |
| Race | | | | | | | | | | | | | | |
| White | – | – | 0.03 | 2.67 | – | – | 0.02 | 2.25 | – | – | – | – | – | – |
| Black/African American | – | – | a | a | – | – | 0.03 | 2.32 | – | – | – | – | – | – |
| Asian/Other/Multiple | – | – | 0.02 | 2.63 | – | – | 0.02 | 2.63 | – | – | – | – | – | – |
| Ethnicity | | | | | | | | | | | | | | |
| Not Hispanic or Latino | – | – | 0.03 | 2.69 | – | – | 0.03 | 2.34 | – | – | – | – | – | – |
| Hispanic/Latino | – | – | 0.03 | 2.27 | – | – | 0.02 | 2.27 | – | – | – | – | – | – |
| **1-hydroxypyrene (1-OH-PYR)** | | | | | | | | | | | | | | |
| Overall | 0.13 | 2.59 | – | – | – | – | 0.06 | 14.3 | – | – | – | – | – | – |
| Income (USD, k = 1,000) | | | | | | | | | | | | | | |
| < 15k (< 20k GAPPS) | 0.18 | 2.11 | – | – | – | – | 0.12 | 9.01 | – | – | – | – | – | – |
| 15k–25k (20k–30k GAPPS) | 0.17 | 2.35 | – | – | – | – | 0.07 | 20.5 | – | – | – | – | – | – |
| 25k–45k (30k–50k GAPPS) | 0.14 | 2.28 | – | – | – | – | 0.09 | 10.8 | – | – | – | – | – | – |
| 45k–55k (50k–60k GAPPS) | 0.09 | 2.70 | – | – | – | – | a | a | – | – | – | – | – | – |
| 55k–65k (60k–70k GAPPS) | 0.10 | 2.31 | – | – | – | – | 0.10 | 4.83 | – | – | – | – | – | – |
| 65k–75k (70k–80k GAPPS) | 0.09 | 3.49 | – | – | – | – | 0.02 | 59.5 | – | – | – | – | – | – |
| ≥ 75k (80k GAPPS) | 0.07 | 2.99 | – | – | – | – | 0.02 | 18.4 | – | – | – | – | – | – |
| Education | | | | | | | | | | | – | – | – | – |
| < High school | 0.18 | 2.23 | – | – | – | – | 0.11 | 10.27 | – | – | – | – | – | – |
| High school completion | 0.17 | 2.13 | – | – | – | – | 0.10 | 10.33 | – | – | – | – | – | – |
| Graduated college/technical school | 0.11 | 2.75 | – | – | – | – | 0.02 | 39.9 | – | – | – | – | – | – |
| Some graduate work or graduate/professional degree | 0.06 | 3.36 | – | – | – | – | 0.03 | 12.51 | – | – | – | – | – | – |
| Race | | | | | | | | | | | | | | |
| White | 0.09 | 2.37 | – | – | – | – | 0.04 | 16.5 | – | – | – | – | – | – |

(*Continued*)

**Table 3.** (Continued)

| Socioeconomic characteristic | CANDLE (Memphis) | | TIDES (San Francisco) | | TIDES (Minneapolis) | | TIDES (Rochester) | | TIDES (Seattle) | | GAPPS (Seattle) | | GAPPS (Yakima) | |
|---|---|---|---|---|---|---|---|---|---|---|---|---|---|---|
| | GM (ng/mL) | GSD | GM (ng/mL) | GSD | GM (ng/mL) | GSD | GM (ng/mL) | GSD | GM (ng/mL) | GSD | GM (ng/mL) | GSD | GM (ng/mL) | GSD |
| Black/African American | 0.17 | 2.21 | – | – | – | – | 0.15 | 7.8 | – | – | – | – | – | – |
| Asian/Other/Multiple | 0.12 | 2.28 | – | – | – | – | 0.04 | 18.8 | – | – | – | – | – | – |
| Ethnicity | | | | | | | | | | | | | | |
| Not Hispanic or Latino | 0.14 | 2.34 | – | – | – | – | 0.07 | 13.02 | – | – | – | – | – | – |
| Hispanic/Latino | 0.20 | 2.39 | – | – | – | – | 0.05 | 16.36 | – | – | – | – | – | – |

GM, geometric mean. GSD, geometric standard deviation.

[a]Not calculated because of low cell count (n ≤ 5).

most sites (with the exception of TIDES Minneapolis and TIDES Rochester), 1-hydroxynaphthalene was the next highest concentration detected. The CANDLE (Memphis) GMs for these metabolites (1- and 2-hydroxynaphthalene), as well as 2/3/9-hydroxyfluorene, and 1/9-hydroxyphenanthrene, were greater than observed in other study sites.

Fig 2 shows the within-site correlations between OH-PAH concentrations by site. Within all sites, at least two of the phenanthrene metabolites (1/9-, 2-, 3- and/or 4-hydroxyphenanthrene) were strongly correlated (rho > 0.7).

## Repeated OH-PAH measures across pregnancy in the TIDES Cohort

Participants (n = 677) from the four TIDES study sites provided at least two prenatal urine samples. Among them, the median gestational age was 11 weeks at the early pregnancy visit (range: 5–23 weeks), 21 weeks at the mid pregnancy visit (range: 13–35 weeks), and 32 weeks at the late pregnancy visit (range: 26–41 weeks) (Table 1). We did not observe any obvious or consistent trends by pregnancy timing for any OH-PAH at any of the four TIDES sites. For OH-PAHs with high detection frequencies for all study sites, we estimated intra-site

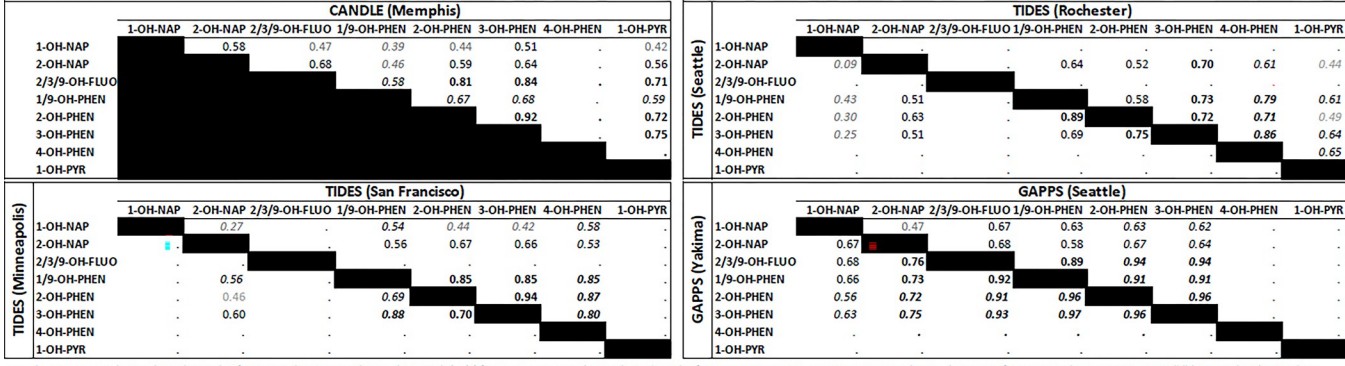

**Fig 2. Correlation coefficients of log-transformed concentrations (ng/mL) of 8 OH-PAHs among non-smoking individuals, by study site (n = 1,695).** Gray font indicates no correlation (rho <0.5); regular font indicates moderate correlation (rho 0.5–0.7); bold font indicates strong correlation (rho > 0.7); italic font suggests cautious interpretation due to detection frequencies 60–85%; missing (".") indicates low confidence due to detection frequencies <60%. Analyte abbreviations: 1-OH-NAP, 1-hydroxynaphthalene; 2-OH-NAP, 2-hydroxynaphthalene; 2/3/9-OH-FLUO, 2/3/9-hydroxyfluorene; 1/9-OH-PHEN, 1/9-hydroxyphenanthrene; 2-OH-PHEN, 2-hydroxyphenanthrene; 3-OH-PHEN, 3-hydroxyphenanthrene; 4-OH-PHEN, 4-hydroxyphenanthrene; 1-OH-PYR, 1-hydroxypyrene.

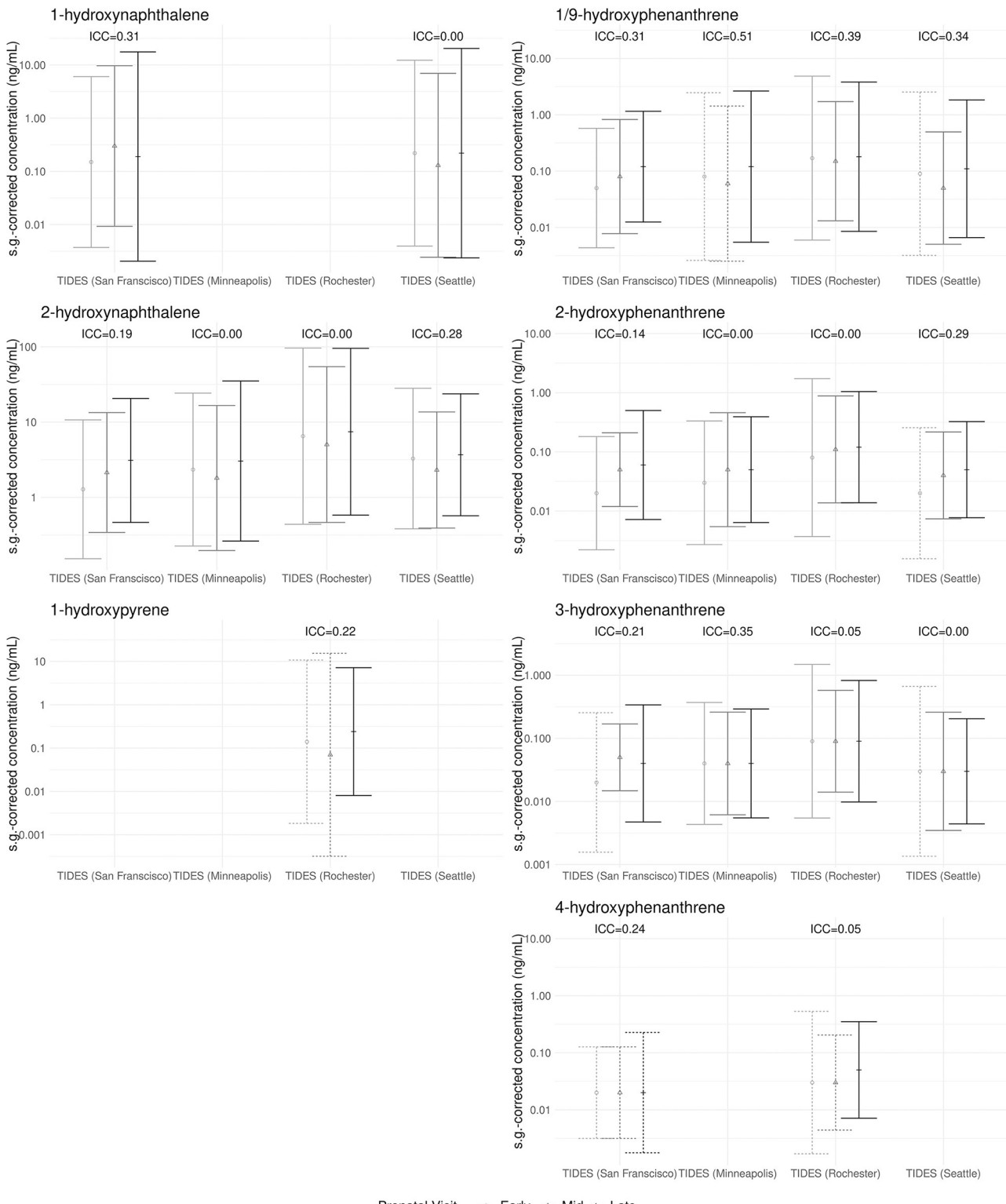

**Fig 3. Geometric mean (95% confidence interval) of s.g.-corrected concentrations (ng/mL) of 7 OH-PAHs repeatedly measured among non-smoking individuals across pregnancy, by study site (TIDES only, n = 677 with at least two observations).** Intra-site ICCs are shown for metabolites with > 95% detection frequencies at all four TIDES sites. Analytes with detection frequencies < 60% are not shown; dashed lines indicate 60–85% detection frequencies, interpret with caution; solid lines indicate > 85% detection frequencies, very confident. s.g., specific gravity.

ICCs ≤ 0.51 (Fig 3). ICCs for the pooled TIDES sample were as follows: 0.47 for 2-hydroxy-naphthalene and 0 for 2- and 3-hydroxyphenanthrene.

## Socioeconomic characteristics

Table 3 shows s.g.-corrected GM OH-PAH concentrations stratified by site and socioeconomic characteristics. S3–S7 Tables show results from the Tobit regressions of log OH-PAH concentrations on income, education, maternal age, race, and ethnicity, respectively, adjusted for s.g.. Income was not consistently associated with OH-PAH levels for any analyte or site except for GAPPS Seattle, where levels increased with household size adjusted household income, although the small sample size of the complete case analysis (n = 51) limits interpretation (S3 Table). Among CANDLE and TIDES Rochester participants, OH-PAHs were lower in the higher two education categories compared to "< High school/high school completion" but this trend was not consistently observed among participants at the other sites (S4 Table). Age was not consistently associated with OH-PAH levels for any site except for CANDLE where each year of age was associated with statistically significantly lower levels of 1- and 2-hydroxynaphthlane, 2/3/9-hydroxyfluorene, 3-hydroxyphenanthrene, and 1-hydroxypyrene (S5 Table). Race was not associated with OH-PAHs at most sites except CANDLE and TIDES Rochester. In CANDLE, 1- and 2-hydroxynaphthalene, 2/3/9-hydroxyfluorene, 2- and 3-hydroxyphenanthrene was higher in Black/African American vs. White participants, and 1-hydroxypyrene was higher in both Black/African American and Other participants compared to White participants. In TIDES Rochester, 2-hydroxynaphthalene was higher in both Black/African American and Other participants compared to White participants, and 2- and 3-hydroxyphenanthrene and 1-hydroxypyrene were higher in Black/African American vs. White participants (Tables 3 and S6). Ethnicity was not consistently associated with OH-PAHs at any site (Tables 3 and S7).

## Discussion

In this descriptive analysis of pregnant individuals recruited from 7 different community-based cohort study sites in the United States, we observed ubiquitous PAH exposures across all sites. We excluded individuals who smoked during pregnancy from our geometric mean calculations given the established contribution of tobacco smoke exposure to urinary OH-PAH concentrations [1]. Low molecular weight (LMW) PAHs such as naphthalene, fluorene, phenanthrene are detected in abundance in tobacco cigarette yields, whereas higher molecular weight PAHs are less so [36]. We observed expected differences in median concentrations of naphthalene, fluorene and pyrene metabolites between smokers and non-smokers in two study samples (CANDLE and TIDES Rochester). These smoker vs. non-smoker differences were not as apparent in the phenanthrene metabolite medians at any site, or in the naphthalene and fluorene metabolite medians at either GAPPS site, indicating that sources other than active smoking were influential. In females aged 16–46 years in the NHANES 2011–2012, median levels of all the target analytes were higher in smokers compared to non-smokers, so we expected to find similar differences in the PATHWAYS cohorts.

Among nonsmokers, the distribution of urinary OH-PAH concentrations varied by site. Among the 8 OH-PAHs analyzed, 2-hydroxynaphthalene was detected in the highest concentrations at all sites, and we observed the highest average concentrations of four metabolites (1- and 2- hydroxynaphthalene, 2/3/9-hydroxyfluorene, and 1/9-hydroxyphenanthrene) at the CANDLE site. Differences in indoor exposures could help explain why we saw the highest average concentrations of most metabolites at the Memphis site; however, outdoor sources and diet may also play roles [37–39]. Indeed, differences in indoor and outdoor sources and

diet, which may further be patterned by demographic characteristics that differ between the cohorts, may help to explain differences in observed concentrations between the other sites. We expected 2-hydroxynaphthalene to be detected in the highest concentrations given naphthalene is the most volatile of the four parent PAHs we studied and accounted for ~87% of the total concentration of 16 PAHs measured in the U.S. Environmental Protection Agency's (EPA) Urban Air Toxic Monitoring program from 1990 to 2014 [40]. In addition to its many combustion-related outdoor sources, naphthalene can also be emitted indoors from mothballs and older moth repellant formulations, as well as from building materials such as caulking, carpet pads, and flooring [37]. For the LMW PAHs (i.e., 2–3 rings) we studied—naphthalene, fluorene, and phenanthrene—indoor air may dominate outdoor air and diet as exposure sources for the average American [37,41]. Shin et al. (2013) compared daily PAH intake estimated from urinary OH-PAH concentrations in NHANES 2001–2002 with daily intakes modeled using county-level emissions data from EPA's National Air Toxics Assessment (NATA), indoor air concentrations from small panel studies, and the CalTOX multimedia exposure model to estimate PAH intakes from food [37]. In this study, the distributions of biomarker-based intakes most closely resembled the distributions of modeled indoor inhalation intakes for the LMW PAHs (naphthalene, fluorene, phenanthrene) and for pyrene, which with 4 rings is considered a high molecular weight (HMW) PAH but with a molecular weight just above 200 g/mol, may behave similarly to the LMW PAHs [37].

## Comparisons with other studies of pregnant individuals

Three other published studies measured urinary OH-PAH levels in mostly non-smoking pregnant individuals in community-based cohorts [31,42,43]. Cathey, et al. measured 8 OH-PAHs in a cohort in Puerto Rico (n = 50) and another in Boston (n = 200) [31]. Nethery, et al. measured 10 OH-PAHs in a cohort of 19 participants in Hamilton, ON, Canada [42]. Polańksa, et al. described PAH exposure in pregnant individuals from two study sites in Poland (n = 210) but reported only phenanthrene and pyrene metabolites [41,44].

 Several of our observations are consistent with the smaller studies of pregnant individuals described above. These studies also detected OH-PAHs in a high proportion of urine samples [31,41,42]. The two that analyzed a wider range of metabolites also observed 1- and 2-hydroxynaphthalene in the highest concentrations [31,42]. Cathey, et al. observed significant differences in the concentrations of certain metabolites between their two study sites: 1-hydroxynaphthalene, 2-hydroxyfluorene, 1-hydroxyphenanthrene (higher in participants from Boston) and 2-hydroxynaphthalene (higher in participants from Puerto Rico) [31]. Both our study and Cathey, et al. analyzed 1- and 2-hydroxynapthalene, 4-hydroxyphenanthrene, and 1-hydroxyprene. We observed a range of uncorrected GM concentrations in the PATHWAYS data (among sites with ≥ 65% detection frequency) that included those observed in the Boston and Puerto Rico study samples with two exceptions: 2-hydroxynapthalene and 1-hydroxyprene GMs were higher in Puerto Rico than Boston and all PATHWAYS samples [31].

## Comparison with females of reproductive age in NHANES 2011–2012 data

The NHANES 2011–2012 data showed uncorrected median OH-PAH concentrations within the range of median values observed across all PATHWAYS study sites for all metabolites except 1- and 2-hydroxynapthalene where NHANES medians were slightly above the range of median values observed across PATHWAYS sites. Woodruff, et al. described urinary OH-PAH levels among pregnant participants (n = 84–91) in NHANES 2003–2004, 9% of whom were smokers [45]. Their median uncorrected OH-PAH concentrations were generally

like those in the NHANES 2011–2012 data we examined (i.e., n = 535 non-smoking females of reproductive age) except that the 2-hydroxynaphthalene median was notably higher in the 2011–2012 subsample (4.9 ng/mL) compared to the 2003–2004 pregnant subsample (2.4 ng/mL).

## Repeated measures

We did not observe consistent trends based on timing of gestation in the repeated measures of OH-PAHs across pregnancy in the TIDES cohort. The low ICCs we observed across the repeated measures within each TIDES site may illustrate the limitations of using a single spot urine measurement to represent typical PAH exposures but may also be related to the potential influence of season relative to pregnancy stage, to additional variability introduced by changes in glomerular filtration rate and/or other physiological processes across pregnancy [46], or to some combination of factors. Other studies that have also looked at repeated measures of OH-PAHs in pregnancy have not observed significant differences or strong trends across pregnancy [31,41,42]. A study of non-pregnant females of reproductive age similarly observed low to moderate ICCs for OH-PAHs (0.23–0.67), suggesting the potential importance of analyzing multiple samples within participants for association studies [47].

## Socioeconomic characteristics

Among the PATHWAYS participants who did not smoke during pregnancy, we observed inconsistent patterns in income, education, and race related to OH-PAH concentrations. Other studies have suggested level of education may be inversely correlated with OH-PAH concentrations in pregnant people [31]. It is important that subsequent analyses investigating potential environmental sources of exposure rely on multivariable models and include these important factors as populations may be differentially exposed to PAHs based on socioeconomic characteristics.

## Strengths and limitations

Strengths of this study were the large sample size, geographic and socioeconomic diversity, and participation of pregnant individuals from seven U.S. study sites. Additionally, we analyzed OH-PAHs in urine which captures aggregate PAH exposures, instead of focusing on a single medium such as air or diet. For four of the PATHWAYS study sites, we evaluate repeated measures of OH-PAHs during pregnancy. By using an MLE approach, validated with visual diagnostics, to impute values below the detection limit, we generated less biased geometric mean estimates compared to comparable studies using single value (i.e., LOD, LOD/sqrt 2, ½ LOD) imputation [10–12,15,25–27]. This in turn allowed us to make valid comparisons of average concentrations across sites despite site-specific differences in analytical detection limits. Appropriate handling of left-censored observations is especially important for large multi-site studies like ECHO PATHWAYS where detection limits can vary by a factor of 10 or more depending on site. Although we presented data from different study sites, it is important to consider in the interpretation of the data that, with the exception of the Memphis site, which represents Shelby County, Tennessee, the study samples are not statistically representative of the regional population in the area where they were recruited. It is also important to note that the results we presented are descriptive in nature, based on one urine sample. For compounds metabolized in hours vs. days or months, a single spot urine sample does not capture exposures across all of pregnancy. Additionally, our GM estimates did not account for seasonal variation, indoor or outdoor sources, diet, or other factors related to PAH exposure. These will be the focus of future investigations planned to examine associations in subsequent multivariable analyses.

## Conclusion

This descriptive analysis built on an existing limited literature on PAH exposure during pregnancy using sensitive and accurate measurements of urinary metabolite biomarkers and utilized a less-biased approach for handling observations below the detection limit compared to single-value imputation. Our overall sample size was comprised of nearly 2,000 pregnant individuals who were non-smokers during pregnancy. We had repeated urinary PAH measurements during pregnancy for nearly 700 participants from four different study sites. We characterized PAH exposure based on 8 metabolites by socioeconomic characteristics in 7 study samples.

In this 7-site analysis, although PAH exposure was ubiquitous at all sites, exposure patterns varied by study site and PAH metabolite. These findings highlight the importance of better understanding PAH sources and their pediatric health outcomes attributed to early life PAH exposure. Future studies will examine the associations between OH-PAHs and potential environmental sources of exposure using multivariable models, controlling for the influence of study sample characteristics such as region, season and demographic factors.

## Supporting information

**S1 Fig. Participant flow diagram by cohort and site.**
(PDF)

**S2 Fig. MLE fitted distributions (dashed line), histograms, estimated modes (green vertical line), and analytical detection limits (blue vertical line) of log-transformed uncorrected (S2A Fig) and specific gravity-corrected (S2B Fig) urinary 1-hydroxypyrene among CANDLE participants.** Upper panels (salmon) are non-smokers; lower panels (turquoise) are smokers. 1-hydroxypyrene detection frequencies were 89% for non-smokers (n = 885) and 97% for smokers (n = 103).
(PDF)

**S3 Fig. MLE fitted distributions (dashed line), histograms, estimated modes (green vertical line), and analytical detection limits (blue vertical line) of log-transformed uncorrected (S3A Fig) and specific gravity-corrected (S3B Fig) urinary 2-hydroxyphenanthrene among GAPPS (Seattle) participants.** Upper panels (salmon) are non-smokers; lower panels (turquoise) are smokers. 2-hydroxyphenanthrene detection frequencies were 65% for non-smokers (n = 91) and 62% for smokers (n = 21).
(PDF)

**S1 Table. Accuracy and precision of the CANDLE and GAPPS mid-pregnancy urinary PAH metabolite measurements.**
(XLSX)

**S2 Table. Accuracy and precision of the TIDES urinary PAH metabolite measurements by analytical batch; note that each batch contains samples from all three pregnancy trimesters.**
(XLSX)

**S3 Table. Income coefficients from Tobit regressions of log urinary OH-PAH concentration vs. urinary specific gravity and household count adjusted household income, by cohort and site, non-smokers, complete cases (analytes detected in < 60% of samples not shown).**
(XLSX)

**S4 Table. Education category coefficients from Tobit regressions of log urinary OH-PAH concentration vs. urinary specific gravity and highest education level, by cohort and site, non-smokers, complete cases (analytes detected in $<$ 60% of samples not shown).**
(XLSX)

**S5 Table. Maternal age coefficients from Tobit regressions of log urinary OH-PAH concentrations vs. urinary specific gravity and continuous age (years), by cohort and site, non-smokers, complete cases (analytes detected in $<$ 60% of samples not shown).**
(XLSX)

**S6 Table. *P*-values of the maternal race coefficient in Tobit regressions of urinary OH-PAH concentrations vs. urinary specific gravity and race, by cohort and site, non-smokers.**
(XLSX)

**S7 Table. *P*-values of the maternal ethnicity coefficient in Tobit regressions of urinary OH-PAH concentrations vs. urinary specific gravity and ethnicity, by cohort and site, non-smokers.**
(XLSX)

## Acknowledgments

The authors thank the participating families for their time and commitment to the CANDLE, TIDES, and GAPPS. We also appreciate the dedication of the study research staff and investigators who have worked diligently independently and collaboratively to generate the ECHO PATHWAYS Consortium. Special thanks to Tomomi Workman of the University of Washington for expert graphics assistance.

## Author Contributions

**Conceptualization:** Erin E. Masterson, Christine T. Loftus, Shanna Swan, Nicole R. Bush, Sheela Sathyanarayana, Kaja Z. LeWinn, Catherine J. Karr.

**Data curation:** Erin E. Masterson, Christine T. Loftus, Revathi Muralidharan.

**Formal analysis:** Erin E. Masterson, Anne M. Riederer, Revathi Muralidharan, Kurunthachalam Kannan, Morgan Robinson.

**Funding acquisition:** Leonardo Trasande, Emily S. Barrett, Ruby H. N. Nguyen, Shanna Swan, W. Alex Mason, Nicole R. Bush, Sheela Sathyanarayana, Kaja Z. LeWinn, Catherine J. Karr.

**Investigation:** Erin E. Masterson, Ruby H. N. Nguyen, Kurunthachalam Kannan.

**Methodology:** Anne M. Riederer, Adam A. Szpiro.

**Project administration:** Erin E. Masterson, Christine T. Loftus, Leonardo Trasande, Emily S. Barrett, Shanna Swan, W. Alex Mason, Nicole R. Bush, Sheela Sathyanarayana, Kaja Z. LeWinn, Catherine J. Karr.

**Resources:** Christine T. Loftus.

**Software:** Erin E. Masterson, Anne M. Riederer.

**Supervision:** Leonardo Trasande, Kurunthachalam Kannan, Shanna Swan, W. Alex Mason, Sheela Sathyanarayana, Kaja Z. LeWinn, Catherine J. Karr.

**Validation:** Anne M. Riederer, Christopher D. Simpson.

**Visualization:** Erin E. Masterson, Anne M. Riederer, Revathi Muralidharan.

**Writing – original draft:** Erin E. Masterson, Anne M. Riederer.

**Writing – review & editing:** Erin E. Masterson, Anne M. Riederer, Christine T. Loftus, Erin R. Wallace, Adam A. Szpiro, Christopher D. Simpson, Emily S. Barrett, Ruby H. N. Nguyen, Kurunthachalam Kannan, Morgan Robinson, Nicole R. Bush, Sheela Sathyanarayana, Kaja Z. LeWinn, Catherine J. Karr.

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
