## [Decision Letter · Decision Letter 0]

27 Sep 2023

PONE-D-23-14326Urinary polycyclic aromatic hydrocarbon (PAH) metabolite concentrations

in three pregnancy cohorts from 7 U.S. study sitesPLOS ONE

Dear Dr. Riederer,

Thank you for submitting your manuscript to PLOS ONE. After careful consideration, we feel that it has merit but does not fully meet PLOS ONE’s publication criteria as it currently stands. Therefore, we invite you to submit a revised version of the manuscript that addresses the points raised during the review process.

We look forward to receiving your revised manuscript.

Kind regards,

Govarthanan Muthusamy

Academic Editor

PLOS ONE

Journal Requirements:

Reviewers' comments:

Reviewer's Responses to Questions

**Comments to the Author**

1. Is the manuscript technically sound, and do the data support the conclusions?

Reviewer #1: Yes

Reviewer #2: Yes

2. Has the statistical analysis been performed appropriately and rigorously? 

Reviewer #1: Yes

Reviewer #2: Yes

3. Have the authors made all data underlying the findings in their manuscript fully available?

Reviewer #1: Yes

Reviewer #2: Yes

4. Is the manuscript presented in an intelligible fashion and written in standard English?

Reviewer #1: Yes

Reviewer #2: Yes

5. Review Comments to the Author

Reviewer #1: I would like to express my gratitude for the opportunity to review your manuscript titled " Urinary polycyclic aromatic hydrocarbon (PAH) metabolite concentrations in three pregnancy cohorts from 7 U.S. study sites". Polycyclic aromatic hydrocarbon (PAH) is a class of chemicals that hold significant environmental importance. It is intriguing to see an updated study on the between urinary PAH and pregnancy cohort. The study is well written. Consequently, I have some questions and suggestions that need to be addressed:

Abstract: The abstract could be organized using subheadings such as "Objective," "Methods," "Results," and "Conclusion."

Methods: Since this study was used to data set, I assume it involved multiple data sets. It is highly recommended that you clarify how many cycles' data sets were merged and provide information on the sample size of each data set. Additionally, please add the flow chart accordingly to reflect these details.

Urinary Polyaromatic Hydrocarbon: Please provide information on how the samples were stored and at what temperature.

Discussion section: The discussion should emphasize the novelty of your research results by comparing them with findings from previous studies. It would be beneficial to delve deeper into the interpretation of the results and provide a more comprehensive analysis.

There are few studies on same population and it is recommended to cite the following studies in this study: https://www.sciencedirect.com/science/article/abs/pii/S0013935122001025, https://academic.oup.com/inthealth/article/15/2/161/6617857, https://link.springer.com/article/10.1007/s10653-022-01438-y.

Overall, the manuscript would benefit from improvements in the English language. I recommend having it edited the language quality with professional standards.

Reviewer #2: Urinary polycyclic aromatic hydrocarbon (PAH) metabolite concentrations

in three pregnancy cohorts from 7 U.S. study sites

D-23-14326

Masterson et al.

Masterson et al. measured and analyzed 12 PAH metabolites in the urine of pregnant women living in the United States from seven metropolitan sites in the ECHO PATHWAYS Consortium.

The data collected in this study can be useful for evaluating the relationship between prenatal exposure to PAH exposure and health outcomes and evaluating sources of exposure (e.g., planned studies). However, without a more detailed analysis of the exposure data, it is unclear how this study per se advances the science, as urinary data on women of reproductive age is available from NHANEs. [this study doesn’t can’t evaluate differences due to pregnancy as samples collected before pregnancy are not available}.

From my perspective, the study would be more informative and public health-relevant if it conducted a deeper dive to explore differences in urinary PAH levels.

Some suggestions are as follows.

• SES/Education: conduct a formal analysis (rather than just providing data stratified by SES) to determine the relationship between SES and urinary PAH metabolite levels

• Age: conduct a formal analysis to determine the relationship been age and urinary PAH metabolite levels.

• Results should note differences in cohorts by age (e.g., women in SF and Seattle, Memphis, and Rochester are younger).

• Race/Ethnicity: Provide the racial/ethnicity group in Table 1, stratify PAH levels by race/ethnicity, and conduct a formal analysis.

• Geographical location: Are any participants from rural or suburban areas outside the city? If so, this could be explored.

• Consider summing the metabolites for the analyses by age, SES, race/ethnicity, and possibly geographical location.

Minor comments

• Figure 2: format the graph for Candles to be consistent with the other sites.

• Repeated measures across pregnancy. How would changes in the Glomerular filtration rate during pregnancy affect urinary concentrations of PAHs metabolites?

6. PLOS authors have the option to publish the peer review history of their article (what does this mean?). If published, this will include your full peer review and any attached files.

Reviewer #1: **Yes: **Manthar Ali Mallah

Reviewer #2: No

---

## [Author Response · Author response to Decision Letter 0]

14 Dec 2023

Thank you so much for the thoughtful comments and suggestions. Below are our responses-- kindly note that there are some additional tables we prepared that do not show up below but are included in the Reviewer Response letter we uploaded. Thank you. 

Masterson et al. - Response to Reviewers

Reviewer #1: I would like to express my gratitude for the opportunity to review your manuscript titled " Urinary polycyclic aromatic hydrocarbon (PAH) metabolite concentrations in three pregnancy cohorts from 7 U.S. study sites". Polycyclic aromatic hydrocarbon (PAH) is a class of chemicals that hold significant environmental importance. It is intriguing to see an updated study on the between urinary PAH and pregnancy cohort. The study is well written. Consequently, I have some questions and suggestions that need to be addressed:

Abstract: The abstract could be organized using subheadings such as "Objective," "Methods," "Results," and "Conclusion."

Thank you for the recommendation-- we added the suggested subheadings to the abstract. 

Methods: Since this study was used to data set, I assume it involved multiple data sets. It is highly recommended that you clarify how many cycles' data sets were merged and provide information on the sample size of each data set. Additionally, please add the flow chart accordingly to reflect these details.

Thank you for the suggestion. We made a flow chart and added it to the manuscript as Figure S1 in the Supporting Information. We also added this sentence to the Participants section of Materials and Methods:

"Figure S1 in the Supporting Information shows the participant flow diagram by cohort and site."

"Urinary Polyaromatic Hydrocarbon: Please provide information on how the samples were stored and at what temperature."

Thank you for catching this. We added the following sentence to the beginning of the OH-PAH Methods section: 

"Samples were stored at -80°C in each cohort's respective biorepository until analysis for OH-PAHs at the New York State Department of Health, Wadsworth Laboratory."

Discussion section: The discussion should emphasize the novelty of your research results by comparing them with findings from previous studies. It would be beneficial to delve deeper into the interpretation of the results and provide a more comprehensive analysis.

Thank you for this constructive feedback-- please see related comment and our response below. 

There are few studies on same population and it is recommended to cite the following studies in this study: https://www.sciencedirect.com/science/article/abs/pii/S0013935122001025, https://academic.oup.com/inthealth/article/15/2/161/6617857, https://link.springer.com/article/10.1007/s10653-022-01438-y.

Thank you for the suggestions. We added the recommended citations to a new paragraph we added to Statistical Methods (see related comment below): "Since income and education were previously associated with urinary OH-PAH concentrations in pregnant (Cathey et al. 2018) and non-pregnant people (Mallah et al. 2023a, 2023b; Mallah et al. 2022; Liu and Jia 2016), we compared GM/GSD OH-PAH concentrations across income and education categories..."

Overall, the manuscript would benefit from improvements in the English language. I recommend having it edited the language quality with professional standards.

Thank you for the feedback . Senior scientists from our research group who are native English speakers with extensive publication records, have carefully reviewed this revised version of our manuscript for language use. 

Reviewer #2...Masterson et al. measured and analyzed 12 PAH metabolites in the urine of pregnant women living in the United States from seven metropolitan sites in the ECHO PATHWAYS Consortium.

The data collected in this study can be useful for evaluating the relationship between prenatal exposure to PAH exposure and health outcomes and evaluating sources of exposure (e.g., planned studies). However, without a more detailed analysis of the exposure data, it is unclear how this study per se advances the science, as urinary data on women of reproductive age is available from NHANEs. [this study doesn’t can’t evaluate differences due to pregnancy as samples collected before pregnancy are not available}.

From my perspective, the study would be more informative and public health-relevant if it conducted a deeper dive to explore differences in urinary PAH levels.

Thank you for this constructive feedback. The purpose of this study is to describe and characterize OH-PAH concentrations in pregnancy among participants in the large ECHO/PATHWAYS consortium. This data is a unique characterization of urinary PAH exposure during pregnancy given few women in the NHANES database were pregnant during sampling times. We are currently preparing a separate manuscript reporting results of analyses emphasizing sources of exposure where we formally test the hypothesis that exposure to environmental tobacco smoke, measured via urinary cotinine, is associated with urinary OH-PAHs in the pooled data. The new models are adjusted for specific gravity, site, season, year, analytical batch, and body mass index, and for SES characteristics such as household income, education, employment status, and neighborhood deprivation index. In similarly-adjusted models, we are also formally testing the hypothesis that ambient air pollution, estimated via 24-hr average PM2.5 concentrations at participants' residential census tracts (available from CDC's Environmental Public Health Tracking Network), is associated with urinary OH-PAHs. 

We prefer to keep the present manuscript descriptive in nature, so that we can present a detailed picture of OH-PAH levels in this large study sample. We believe this will be useful to others examining OH-PAH concentrations among pregnant people in the United States.

To satisfy the reviewer's curiosity about race/ethnicity patterns in the present analyses, however, we conducted post hoc exploratory race/ethnicity analyses -- see related response below. 

Some suggestions are as follows.

• SES/Education: conduct a formal analysis (rather than just providing data stratified by SES) to determine the relationship between SES and urinary PAH metabolite levels

Thank you for the suggestion. There is indeed evidence in the literature that both income and education are associated with urinary OH-PAH levels in both pregnant and non-pregnant people. This was the reason we chose to highlight these characteristics in Table 3. In response to the suggestion, we conducted additional analyses of urinary OH-PAH concentrations by income, and separately, education category, by cohort and site, using Tobit regression to account for left censoring in the data and adjusting for urinary specific gravity. 

We added the following new paragraph describing these analyses to Statistical Methods: 

"Since income and education were previously associated with urinary OH-PAH concentrations in pregnant (Cathey et al. 2018) and non-pregnant people (Mallah et al. 2023a, 2023b; Mallah et al. 2022; Liu and Jia 2016), we compared GM/GSD OH-PAH concentrations across income and education categories. We also used Tobit regression to formally evaluate associations between income and education and log-transformed OH-PAH concentrations, accounting for left censoring in the data and adjusting for urinary specific gravity. In these regressions, income was modeled following the recommended approach for multi-site PATHWAYS analyses: harmonized household size (2-3, 4, 5, ≥ 6 people) and income (adjusted for region and inflation to 2012 $USD) from the 4-6 year old visit instead of the prenatal visit because of the proportion of right-censored prenatal TIDES and GAPPS observations, which was addressed at the 4-6 year old visit with additional upper income categories. The models included natural log household income interacted with household size ('household size adjusted household income'), and the corresponding main effects. For the education regressions, the "< High school" and "High school completion" categories were combined to avoid low cell counts in the "< High school" category. Complete case analysis was used for the Tobit regressions."

We also changed "...(including annual household income, ..." to "...(including annual household income at enrollment, ..." the Cohort characteristics section of Methods to make it clear that the descriptive statistics shown in Table 1 apply to household income at enrollment (and not the 4-6 year old visit). 

We also changed the text of the Socioeconomic characteristics section of Results from: 

"Table 3 shows s.g.-corrected GM OH-PAH concentrations stratified by site and socioeconomic characteristics (income and education). We did not formally test the following observations. We observed some trends between lower concentrations of most OH-PAHs and higher income and higher education levels. These varied by site and metabolite. Most consistently, we observed lower concentrations of 2-hydroxynaphthalene among those with higher education levels at all sites. By contrast, OH-PAH concentrations did seem to vary by income at some sites but consistent patterns were not apparent..."

to:

"Table 3 shows s.g.-corrected GM OH-PAH concentrations stratified by site and socioeconomic characteristics (income at enrollment and education). Tables S3 and S4 in the Supporting Information show results from the Tobit regressions of log OH-PAH concentrations on income and education, respectively, adjusted for s.g.. In these analyses, income was not consistently associated with OH-PAH concentration for any analyte or site except for GAPPS Seattle, where concentrations increased with household size adjusted household income, although the small sample size of the complete case analysis (n = 51) limits interpretation (Table S3). Among CANDLE and TIDES Rochester participants, OH-PAHs were lower in the higher two education categories compared to "< High school/high school completion" but this trend was not consistently observed among participants at the other sites."

Last, we changed the Socioeconomic characteristics section of Discussion from

" Among the PATHWAYS participants who did not smoke during pregnancy, we observed patterns in income and/or education level related to OH-PAH concentrations. These patterns varied by study site and metabolite but we did not formally test these differences as our goal was to be descriptive, rather than to test group differences."

to:

"Among the PATHWAYS participants who did not smoke during pregnancy, we observed inconsistent patterns in income and/or education level related to OH-PAH concentrations."

• Age: conduct a formal analysis to determine the relationship been age and urinary PAH metabolite levels.

Thank you for the suggestion. We did not find prior published evidence of associations between age and OH-PAH concentrations in pregnant people so we did not formally test age thus prefer to leave it out of the manuscript. In response to this suggestion, however, we conducted post hoc analyses of urinary OH-PAH concentrations by continuous age (years), by cohort and site, using Tobit regression to account for left censoring in the data and adjusting for urinary specific gravity. P-values from these analyses are summarized below: 

• Results should note differences in cohorts by age (e.g., women in SF and Seattle, Memphis, and Rochester are younger).

Thank you for the suggestion. In response, we changed this sentence in Results, "... The mean age was 29 (SD: 6) years, with variation across sites..." to " The mean age was 29 (SD: 6) years, with variation across sites; in general, CANDLE, TIDES Rochester, and GAPPS Yakima participants were younger than participants at other sites."

• Race/Ethnicity: Provide the racial/ethnicity group in Table 1, stratify PAH levels by race/ethnicity, and conduct a formal analysis.

In the present study, as well as the new analyses described above, we did not formally test the social constructs of race/ethnicity because we have no a priori reason to believe that urinary OH-PAHs differ by race/ethnicity in a manner independent from SES, residence location, environmental tobacco smoke exposure, and other factors included in our new analyses in preparation. In the present study, both race and ethnicity were associated with income and education in non-parametric Kruskal-Wallis tests, however not consistently across sites. Theoretically, there could be race/ethnicity differences in dietary PAH intakes, physiological processes influencing PAH toxicokinetics, or other factors, however empirical evidence for these is scant so we do not include race/ethnicity in the new models.

In response to the reviewer's suggestion, we conducted post hoc analyses of urinary OH-PAH concentrations by race (Black/African American [referent category], White, and Other) by cohort and site, using Tobit regression to account for left censoring in the data and adjusting for urinary specific gravity. P-values from these analyses are summarized below: 

We also conducted post hoc analyses of urinary OH-PAH concentrations by ethnicity (Not Hispanic/Latino [referent category], Hispanic/Latino) by cohort and site, using Tobit regression to account for left censoring in the data and adjusting for urinary specific gravity. P-values from these analyses are summarized below; note that these analyses are not corrected for multiple comparisons:

• Geographical location: Are any participants from rural or suburban areas outside the city? If so, this could be explored.

Thank you for this helpful suggestion and we agree that urban vs. rural (or suburban) differences in urinary OH-PAH concentrations is an interesting scientific question because of the potential influence of urban air pollution. In our upcoming manuscript (in preparation) on PAH exposure factors in the ECHO/PATHWAYS consortium, we used census-tract level estimates of ambient particulate air pollution (PM2.5) from CDC's Environmental Public Health Tracking Network to represent PAH exposures via ambient air pollution. To some degree, this analysis captures urban vs. rural differences since some of the census tracts included are more urban and others more rural. On the other hand, other important PAH exposure sources such as environmental tobacco smoke and diet may or may not vary with urban vs. rural residence location. We examine these potential exposure factors as well in the upcoming manuscript.

• Consider summing the metabolites for the analyses by age, SES, race/ethnicity, and possibly geographical location.

Thank you for the suggestion. We know that some researchers sum PAH metabolites but we do not do this in part because the parent PAHs can come from different sources. For example, 1-hydroxynaphthalene is a metabolite of both naphthalene and the ubiquitous insecticide carbaryl, while 2-hydroxynaphthalene is only a metabolite of naphthalene. Because of this, summing the naphthalene metabolites could provide a misleading estimate of naphthalene exposure, since some portion of the sum may be attributable to carbaryl instead of naphthalene. 

 Similarly, we would not sum naphthalene and pyrene metabolites, for example, since the parent compounds are quite different in terms of their physicochemical properties. Naphthalene behaves more like a volatile organic compound and pyrene behaves more like a semi-volatile organic compound in terms of environmental transport and fate. Since physicochemical properties also influence toxicokinetics (e.g., half-life in the human body), a single spot urine sample could theoretically provide a snapshot of naphthalene and pyrene exposures from two different time points, if one compound is metabolized more quickly than the other. 

 Last, the parent PAHs we considered may also have different modes of toxicity. For example, the U.S. Environmental Protection Agency's Integrated Risk Information System (EPA IRIS) lists different toxicological end points supporting the oral reference doses for naphthalene (decreased body weight in males), fluorene (decreased red blood cells, packed cell volume and hemoglobin), and pyrene (kidney effects) (note: there is currently no oral refere

---

## [Decision Letter · Decision Letter 1]

29 Jan 2024

PONE-D-23-14326R1Urinary polycyclic aromatic hydrocarbon (PAH) metabolite concentrations

in three pregnancy cohorts from 7 U.S. study sitesPLOS ONE

Dear Dr. Riederer,

Thank you for submitting your manuscript to PLOS ONE. After careful consideration, we feel that it has merit but does not fully meet PLOS ONE’s publication criteria as it currently stands. Therefore, we invite you to submit a revised version of the manuscript that addresses the points raised during the review process.

We look forward to receiving your revised manuscript.

Kind regards,

Govarthanan Muthusamy

Academic Editor

PLOS ONE

Journal Requirements:

Reviewers' comments:

Reviewer's Responses to Questions

**Comments to the Author**

1. If the authors have adequately addressed your comments raised in a previous round of review and you feel that this manuscript is now acceptable for publication, you may indicate that here to bypass the “Comments to the Author” section, enter your conflict of interest statement in the “Confidential to Editor” section, and submit your "Accept" recommendation.

Reviewer #1: All comments have been addressed

Reviewer #2: (No Response)

2. Is the manuscript technically sound, and do the data support the conclusions?

Reviewer #1: Yes

Reviewer #2: Yes

3. Has the statistical analysis been performed appropriately and rigorously? 

Reviewer #1: Yes

Reviewer #2: Yes

4. Have the authors made all data underlying the findings in their manuscript fully available?

Reviewer #1: Yes

Reviewer #2: Yes

5. Is the manuscript presented in an intelligible fashion and written in standard English?

Reviewer #1: Yes

Reviewer #2: Yes

6. Review Comments to the Author

Reviewer #1: Manuscript is well written and can be accepted for publication. Comments are well addressed and revised manuscript accordingly.

Reviewer #2: While the authors have made some improvements in the revised manuscript, my original comment –the study would be more informative and public health-relevant if the authors conducted a deeper dive to explore differences in urinary PAH levels–remains. The authors state that they prefer to keep this manuscript and are preparing a separate analysis of the exposure data. This new manuscript sounds exciting. The authors should provide a rationale for what is novel about the current study and why it warrants publication.

I appreciate that the authors conducted in response to my comments (e.g., SES, race/ethnicity); however, they only included the SES analyses in the revised manuscript. The analyses by race and age may help improve the utility of the manuscript as there appears to be differences in PAH metabolite concentrations by race in the largest cohort (which the authors state has the higher number of Black women). The authors suggest they may explore this in more detail in their new paper, but then again, what is the novelty of this paper?

I recommend the authors report the racial/distribution in Table 1 and include the age and race/ethnicity/age analysis in this paper. I did not understand why Black/African-American was the referent group. The rationale for not including age in the paper– We did not find prior published evidence of associations between age and OH-PAH concentrations in pregnant people, so we did not formally test age and thus prefer to leave it out of the manuscript– is not adequate to me. The purpose of research is to explore new hypotheses and ideas.

7. PLOS authors have the option to publish the peer review history of their article (what does this mean?). If published, this will include your full peer review and any attached files.

Reviewer #1: **Yes: **Manthar Ali Mallah

Reviewer #2: No

---

## [Author Response · Author response to Decision Letter 1]

18 Mar 2024

Reviewer #2: While the authors have made some improvements in the revised manuscript, my original comment –the study would be more informative and public health-relevant if the authors conducted a deeper dive to explore differences in urinary PAH levels–remains. The authors state that they prefer to keep this manuscript and are preparing a separate analysis of the exposure data. This new manuscript sounds exciting. The authors should provide a rationale for what is novel about the current study and why it warrants publication…

Thank you and we agree the new manuscript (Riederer et al., in preparation) is exciting (because it shows consistent associations between environmental tobacco smoke exposure and urinary OH-PAHs in the non-smoking pregnant ECHO/PATHWAYS participants). 

We believe that the present descriptive paper (Masterson et al.) is novel because of its large sample size—the largest study of urinary OH-PAHs in pregnant people to date that we are aware of—its geographic diversity (multiple sites across the USA), and our use of maximum likelihood estimation distribution fitting to impute non-detects instead of a single value (e.g., LOD/sqrt(2)), which can bias estimates of central tendency. We thank Reviewer #2 for pushing us to include the exploratory race and ethnicity analyses in the main manuscript which we have now done and hope makes the paper more interesting to readers. Please see our related responses below, and thank you again to Reviewer #2 for such a thoughtful critique.

I appreciate that the authors conducted in response to my comments (e.g., SES, race/ethnicity); however, they only included the SES analyses in the revised manuscript. The analyses by race and age may help improve the utility of the manuscript as there appears to be differences in PAH metabolite concentrations by race in the largest cohort (which the authors state has the higher number of Black women). The authors suggest they may explore this in more detail in their new paper, but then again, what is the novelty of this paper?

I recommend the authors report the racial/distribution in Table 1 and include the age and race/ethnicity/age analysis in this paper. I did not understand why Black/African-American was the referent group…

Thank you for the recommendation; we added the race and ethnicity distributions to Table 1 (with “White” as the referent group, since it was the largest group). 

We also moved the exploratory race and ethnicity analyses into the paper, specifically: 

• We added “race and ethnicity” to the “Cohort characteristics” section of Methods and added a new sentence to the “Statistical analyses” section of Methods: “We also explored associations with…race (White, Black/African American, and an “Other” category comprised of Asian, Native Hawaiian/Pacific Islander, American Indian/Alaska Native, Other, and Multiple to avoid low cell counts), and ethnicity (Not Hispanic/Latino, Hispanic/Latino) using the same approach.”

• We added the geometric means and geometric standard deviations by analyte, site, and race and ethnicity categories to Table 3. 

• We added a new Table S6 (“Maternal age coefficients from Tobit regressions of log urinary OH-PAH concentrations vs. urinary specific gravity and race, by cohort and site, non-smokers, complete cases”) and Table S7 (“P-values of the maternal ethnicity coefficient in Tobit regressions of urinary OH-PAH concentrations vs. urinary specific gravity and ethnicity, by cohort and site, non-smokers”)

• We added a new sentence summarizing the exploratory findings with race and ethnicity to the “Socioeconomic characteristics” section of Results: “Race was not associated with OH-PAHs at most sites except CANDLE and TIDES Rochester. In CANDLE, 1- and 2-hydroxynaphthalene, 2/3/9-hydroxyfluorene, 2- and 3-hydroxyphenanthrene was higher in Black/African American vs. White participants, and 1-hydroxypyrene was higher in both Black/African American and Other participants compared to White participants. In TIDES Rochester, 2-hydroxynaphthalene was higher in both Black/African American and Other participants compared to White participants, and 2- and 3-hydroxyphenanthrene and 1-hydroxypyrene were higher in Black/African American vs. White participants (Table 3; Table S6). Ethnicity was not consistently associated with OH-PAH levels at any site (Table 3; Table S7). ”

• We added race to this sentence in the abstract, “Among non-smoking participants, we observed some patterns by income, education, and race but these were not consistent and varied by site and metabolite.

• We added race to this sentence in the “Socioeconomic characteristics” section of Discussion: “…we observed inconsistent patterns in income, education, and race related to OH-PAH concentrations…”

The rationale for not including age in the paper– We did not find prior published evidence of associations between age and OH-PAH concentrations in pregnant people, so we did not formally test age and thus prefer to leave it out of the manuscript– is not adequate to me. The purpose of research is to explore new hypotheses and ideas.

Thank you for the recommendation—in response, we moved the age analysis into the paper, specifically: 

• we added a new sentence to Methods: “We also explored associations with maternal age in years… using the same approach…”

• we added a new Table S5 (“Maternal age coefficients from Tobit regressions of log urinary OH-PAH concentrations vs. urinary specific gravity and continuous age (years), by cohort and site, non-smokers, complete cases”) 

• we added a new sentence summarizing the exploratory findings with age to the “Socioeconomic characteristics” section of Results: “Age was not consistently associated with OH-PAH levels for any site except for CANDLE where each year of age was associated with statistically significantly lower levels of 1- and 2-hydroxynaphthlane, 2/3/9-hydroxyfluorene, 3-hydroxyphenanthrene, and 1-hydroxypyrene (Table S5).”

---

## [Editor Report · Decision Letter 2]

22 May 2024

Urinary polycyclic aromatic hydrocarbon (PAH) metabolite concentrations

in three pregnancy cohorts from 7 U.S. study sites

PONE-D-23-14326R2

Dear Dr. Riederer,

We’re pleased to inform you that your manuscript has been judged scientifically suitable for publication and will be formally accepted for publication once it meets all outstanding technical requirements.

Kind regards,

Iman Al-Saleh

Academic Editor

PLOS ONE

Additional Editor Comments (optional):

The authors have addressed satisfactorily all comments raised by the reviewers. Thank you.

---

## [Editor Report · Acceptance letter]

4 Jun 2024

PONE-D-23-14326R2 

PLOS ONE

Dear Dr. Riederer, 

I'm pleased to inform you that your manuscript has been deemed suitable for publication in PLOS ONE. Congratulations! Your manuscript is now being handed over to our production team.

Kind regards, 

on behalf of

Dr. Iman Al-Saleh 

Academic Editor

PLOS ONE